# Potentially Optimal Joint Actions Recognition for Cooperative Multi-Agent Reinforcement Learning

**Chang Huang**[1][*]**, Shatong Zhu**[2][*]**, Junqiao Zhao**[1][3][†]**, Hongtu Zhou**[1]**, Di Zhang**[1]**, Hai Zhang**[4]**,
Chen Ye**[1]**, Ziqiao Wang**[1][3]**, Guang Chen**[1][3][5]

[1] School of Computer Science and Technology, Tongji University
[2] Stanford University
[3] MOE Key Lab of Embedded System and Service Computing, Tongji University, Shanghai, China
[4] The University of Hong Kong
[5] Shanghai Innovation Institute
zhaojunqiao@tongji.edu.cn

## Abstract

Value function factorization is widely used in cooperative multi-agent reinforcement learning (MARL). Existing approaches often impose monotonicity constraints between the joint action value and individual action values to enable decentralized execution. However, such constraints limit the expressiveness of value factorization, restricting the range of joint action values that can be represented and hindering the learning of optimal policies. To address this, we propose Potentially Optimal Joint Actions Weighting (POW), a method that ensures optimal policy recovery where existing approximate weighting strategies may fail. POW iteratively identifies potentially optimal joint actions and assigns them higher training weights through a theoretically grounded iterative weighted training process. We prove that this mechanism guarantees recovery of the true optimal policy, overcoming the limitations of prior heuristic weighting strategies. POW is architecture-agnostic and can be seamlessly integrated into existing value factorization algorithms. Extensive experiments on matrix games, difficulty-enhanced predator-prey tasks, SMAC, SMACv2, and a highway-env intersection scenario show that POW substantially improves stability and consistently surpasses state-of-the-art value-based MARL methods.

## 1 Introduction

Multi-agent reinforcement learning (MARL) holds great potential for solving cooperative tasks in domains such as swarm robotics (Huang et al., 2020), autonomous driving (Schmidt et al., 2022), and multi-agent games (Terry et al., 2021). Yet, simultaneous policy learning for multiple agents remains challenging due to non-stationarity and the exponential growth of the joint action space. The centralized training with decentralized execution (CTDE) paradigm has become the standard framework for addressing these challenges, inspiring a wide range of policy-based methods (e.g., MADDPG (Lowe et al., 2017), COMA (Foerster et al., 2018), FOP (Zhang et al., 2021)) and value-based methods (e.g., VDN (Sunehag et al., 2017), QMIX (Rashid et al., 2020a), QPLEX (Wang et al., 2020)).

Among these, QMIX has achieved strong results on benchmarks such as the StarCraft II Multi-Agent Challenge (SMAC) (Samvelyan et al., 2019). QMIX factorizes the joint action-value into individual action-values using a monotonic mixing function, thereby ensuring decentralized execution. However, the monotonicity constraint reduces the expressiveness of the value function, limiting its ability to represent many joint action values and often hindering optimal policy recovery.

---

[*]The first two authors contributed equally to this work.
[†]Corresponding Author

To address this, WQMIX (Rashid et al., 2020b) proposed weighting joint actions during training, ideally emphasizing optimal ones so that the overall joint action-value function ($Q_{tot}$) would approximate the optimal target ($Q^*$). However, identifying the truly optimal joint actions requires traversing the entire joint action space, which is intractable in realistic settings. Practical variants such as CW-QMIX and OW-QMIX replace this exhaustive search with heuristic approximations. Specifically, CW-QMIX anchors its weighting on the $\arg\max Q_{tot}$ rather than the $\arg\max Q^*$, thereby avoiding enumeration of the full action space but introducing inaccuracies. OW-QMIX goes further by directly using $Q_{tot}$ values in an optimistic manner, which amplifies errors when judging whether an action is truly optimal. As a result, both methods misalign the assigned weights with the actual optimal set: suboptimal actions may still receive large weights, while genuinely optimal ones can be under-emphasized. This creates a persistent gap between WQMIX's theoretical promise and its practical realizations.

We propose the **P**otentially **O**ptimal Joint Actions **W**eighting (**POW**) method, which bridges this gap by introducing a recognition-based weighting scheme with provable convergence guarantees. POW employs a recognition module $Q_r$ that explicitly conditions on both state and joint actions, enabling it to identify a set of *potentially optimal joint actions* $\boldsymbol{A}_r$. Training weights are then assigned adaptively, with higher weights for actions in $\boldsymbol{A}_r$. Through iterative updates, we prove that $\boldsymbol{A}_r$ converges to include the true optimal joint actions, ensuring that $Q_{tot}$ aligns its action preferences with those of $Q^*$ without requiring exhaustive search or heuristic approximations. This establishes, for the first time, a consistent link between the theoretical guarantees of weighted value decomposition and its practical implementation.

To validate POW, we instantiate it on top of QMIX (yielding POW-QMIX) and evaluate across diverse benchmarks: matrix games, predator–prey, highway-env, SMAC, and SMACv2. Results show that POW-QMIX outperforms state-of-the-art baselines, particularly in environments with non-monotonic reward structures where existing factorization methods struggle. We further demonstrate that POW can be seamlessly integrated into other value decomposition frameworks, such as VDN and QPLEX, consistently improving their performance. These results highlight both the versatility and scalability of our approach.

In summary, our contributions are:

- We propose POW, a recognition-based joint action weighting framework that provably bridges the gap between the theoretical guarantees of WQMIX and its practical realizations.
- We provide rigorous theoretical analysis, proving that the recognition module ensures convergence of the candidate set $\boldsymbol{A}_r$ toward the true optimal joint actions, thereby enabling optimal policy recovery.
- We conduct extensive experiments across five benchmark families, demonstrating that POW achieves superior performance over strong baselines and generalizes across multiple value factorization architectures.

## 2 PRELIMINARIES

We consider the standard decentralized partially observable Markov decision process (Dec-POMDP) (Oliehoek et al., 2016), defined by a tuple $(\mathcal{S}, \mathcal{A}, P, r, \mathcal{O}, O, n, \gamma)$, where $\mathcal{S}$ is the set of global states, $\mathcal{A} = \times_{i=1}^n \mathcal{A}_i$ the joint action space, $P : \mathcal{S} \times \mathcal{A} \to \Delta(\mathcal{S})$ the transition function, $r : \mathcal{S} \times \mathcal{A} \to \mathbb{R}$ the reward function, $\mathcal{O}$ the set of individual observations, $O : \mathcal{S} \times \{1, \ldots, n\} \to \mathcal{O}$ the observation function, $n$ the number of agents, and $\gamma \in (0, 1)$ the discount factor.

At each timestep $t$, the environment is in state $s_t \in \mathcal{S}$, and agent $i$ selects an action $a_i \in \mathcal{A}_i$ based on its action-observation history $\tau_i \in (\mathcal{O}_i \times \mathcal{A}_i)^*$. The joint action is $\boldsymbol{a} = (a_1, \ldots, a_n)$, leading to the next state $s_{t+1} \sim P(\cdot|s_t, \boldsymbol{a})$ and team reward $r(s_t, \boldsymbol{a})$.

A joint policy $\boldsymbol{\pi} = (\pi_1, \ldots, \pi_n)$ defines each agent's policy $\pi_i$. The objective is to maximize the expected discounted return:

$$J(\boldsymbol{\pi}) = \mathbb{E}_{s_0 \sim \rho, \, \boldsymbol{a}_t \sim \boldsymbol{\pi}, \, s_{t+1} \sim P} \left[ \sum_{t=0}^{\infty} \gamma^t r(s_t, \boldsymbol{a}_t) \right], \tag{1}$$

where $\rho$ is the initial state distribution.

**Centralized Training with Decentralized Execution (CTDE).**   In CTDE, training can leverage global state information, but execution requires each agent to act only on its local trajectory $\tau_i$. This motivates value function factorization methods, where the joint action-value function $Q_{tot}(\tau, \boldsymbol{a})$ is decomposed into individual utilities $Q_i(\tau_i, a_i)$. A common factorization principle is *individual–global–max (IGM)* (Sunehag et al., 2017):

$$\arg\max_{\boldsymbol{a}} Q_{tot}(\tau, \boldsymbol{a}) = \left[ \arg\max_{a_i} Q_i(\tau_i, a_i) \right]_{i=1}^{n}. \tag{2}$$

QMIX (Rashid et al., 2020a) enforces IGM by using a monotonic mixing function:

$$\frac{\partial Q_{tot}}{\partial Q_i} \geq 0, i = 1, ..., n \tag{3}$$

Subsequent works such as QPLEX (Wang et al., 2020) and QTRAN (Son et al., 2019) relax or generalize the decomposition principle, while others (e.g., WQMIX (Rashid et al., 2020b), CW-QMIX, OW-QMIX) assign weights to different joint actions during training. Despite progress, existing approaches either suffer from limited expressiveness (e.g., strict monotonicity) or rely on heuristic weighting schemes that may introduce approximation errors. This motivates our proposed POW framework.

## 3   METHOD

Value decomposition methods under CTDE must factorize the joint action-value function into per-agent utilities. However, with monotonic mixing (e.g., QMIX), an agent may still receive an incorrect penalty if other agents act suboptimally, making it difficult to assign credit to the optimal joint actions. This motivates weighting schemes such as WQMIX (Rashid et al., 2020b), which ideally give higher training weights to optimal joint actions. Yet in practice, variants like CW-QMIX and OW-QMIX must approximate these weights without knowing the true optimal set, introducing errors. POW addresses this challenge by learning a recognition-guided weighting scheme that provably converges toward the optimal set.

### 3.1   ARCHITECTURE OVERVIEW

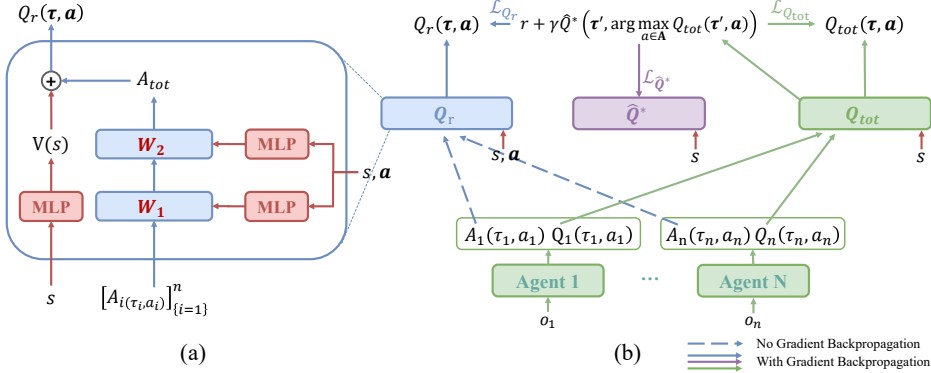

(a)                                                                                         (b)

Figure 1: (a) The $Q_r$ Network structure. (b) The overall architecture of POW method. $Q_{tot}$ can be any value function factorization network satisfying IGM.

Fig. 1 illustrates the POW framework.

Key components are: (1) $\hat{Q}^*$, an unrestricted joint action value estimator that approximates the true optimal action value function $Q^*$ without factorization or monotonic constraints. It provides the bootstrap target shared by all networks during training. (2) $Q_{tot}$, a monotonic mixing network enabling decentralized execution. Its optimality depends on correct weighting of optimal vs. suboptimal joint actions during learning. It can be any value factorization network satisfying IGM

(e.g., QMIX, VDN, QPLEX); (3) the potentially optimal joint actions recognition module $Q_r$. It is used to identify a set of potentially optimal joint actions $\mathcal{A}_r$. $Q_r$ is trained to approximate $\hat{Q}^*$ (or $Q^*$ in theoretical analysis), and its output determines the adaptive training weights applied to each joint action. It takes the joint action as input and provides an expressive joint action value model unconstrained by monotonicity but conforming to IGM.

The architecture of $Q_r$ is shown in Fig. 1 a). The inputs of $Q_r$ include: the global state $s$, the one-hot encoding of joint action, $\boldsymbol{a}$ and the fixed values of individual advantage functions $A_i$. This joint-action conditioning is crucial for distinguishing candidate actions, whereas in QPLEX it primarily increases expressiveness of $Q_{tot}$. By contrast, in POW this design is tied directly to the recognition-weighting mechanism and its convergence properties (see Sec. 3.2). $A_{tot}$ refers to the mixing of the individual agent advantage functions as in QPLEX. The advantage function is defined by $A_i(\tau_i, a_i) = Q_i(\tau_i, a_i) - \max_{a_i \in A_i} Q_i(\tau_i, a_i)$.

$\hat{Q}^*$, $Q_{tot}$ and $Q_r$ (detailed in Sec. 3.2 and Sec. 3.3) share the same Q-learning target:

$$\mathcal{L}_{\hat{Q}^*} = \mathbb{E}[(\hat{Q}^*(\boldsymbol{\tau}, \boldsymbol{a}) - y)^2] \tag{4a}$$

$$\mathcal{L}_{Q_{tot}} = \mathbb{E}[w(s, \boldsymbol{a})(Q_{tot}(\boldsymbol{\tau}, \boldsymbol{a}) - y)^2] \tag{4b}$$

$$\mathcal{L}_{Q_r} = \mathbb{E}[(Q_r(\boldsymbol{\tau}, \boldsymbol{a}) - y)^2] \tag{4c}$$

where

$$y = r + \hat{Q}^*(\boldsymbol{\tau}', \arg\max_{\boldsymbol{a} \in \boldsymbol{A}} Q_{tot}(\boldsymbol{\tau}', \boldsymbol{a})) \tag{5}$$

Together, $\hat{Q}^*$, $Q_{tot}$, and $Q_r$ form a mutually reinforcing system: $Q_r$ proposes potentially optimal actions, the weighting guided by $\mathcal{A}_r$ shapes the update of $Q_{tot}$, and $\hat{Q}^*$ ensures consistent bootstrapping. We later show that this interaction guarantees the convergence of $\mathcal{A}_r$ toward the true optimal action set.

## 3.2 RECOGNITION OF POTENTIALLY OPTIMAL JOINT ACTIONS

We define the recognition module $Q_r$ that explicitly takes as input the global state $s$, individual action-values $Q_i(\tau_i, a_i)$, and the joint action $\boldsymbol{a}$. Here, conditioning on $\boldsymbol{a}$ allows $Q_r$ to assess the value of specific joint actions, enabling recognition of a candidate set $A_r$ of potentially optimal joint actions. Formally, the recognition module is defined as:

$$Q_r(\tau, \boldsymbol{a}) = \sum_{i=1}^{n} \lambda_i(s, \boldsymbol{a})\Big(Q_i(\tau_i, a_i) - \max_{a_i \in A_i} Q_i(\tau_i, a_i)\Big) + V(s), \tag{6}$$

where $\lambda_i(s, \boldsymbol{a}) \geq 0$ are scaling factors. The subtraction term centers each agent's action-value by its best individual choice, while $V(s)$ captures state-dependent value shared across agents. Intuitively, this form highlights whether a joint action sacrifices individual agent optimality, while allowing $Q_r$ to adaptively weight such trade-offs.

Importantly, this construction also guarantees the IGM property. Since the contribution of each agent $i$ is maximized exactly when $a_i$ is its individually optimal action (the centered term becomes zero and all other actions are negative), maximizing $Q_r(\tau, \boldsymbol{a})$ over $\boldsymbol{a}$ is achieved by maximizing each $Q_i(\tau_i, a_i)$ independently. Thus ensuring IGM without enforcing any monotonicity constraint on the underlying $Q_i$.

The training objective of $Q_r$ is to approximate the optimal joint action value function $Q^*$ in theory:

$$\mathcal{L}_{Q_r} = \mathbb{E}\big[(Q_r(\tau, \boldsymbol{a}) - Q^*(\tau, \boldsymbol{a}))^2\big], \tag{7}$$

During training, updates are applied to the parameters of the mixing function, leaving the parameters of the individual action value functions unchanged. The scales $\lambda_i(s, \boldsymbol{a})$ are computed by a hypernetwork, where the global state $s$ and the joint action $\boldsymbol{a}$ are used as inputs to obtain the neural network weights $W_1$ and $W_2$. We take the absolute values of $W_1$ and $W_2$ to ensure that $\lambda_i(s, \boldsymbol{a}) \geq 0$.

**Definition 1** (Potentially optimal joint action set $A_r$). We define $\mathcal{A}_{igm} := \big\{ \boldsymbol{a} \in \mathcal{A} \mid \forall i : a_i \in \arg\max_{a_i \in \mathcal{A}_i} Q_i(\tau_i, a_i) \big\}$ be the set of joint action obtained by greedy individual choices. Let $\hat{\boldsymbol{a}} \in \mathcal{A}_{igm}$, then the potentially optimal joint action set is:

$$A_r := \{ \boldsymbol{a} \in \mathcal{A} \mid Q_r(s, \boldsymbol{a}) \geq Q_r(s, \hat{\boldsymbol{a}}) - C \}, \tag{8}$$

where $C \geq 0$ is a small tolerance constant for stability.

This definition ensures $A_r$ always contains at least the joint greedy action and potentially other promising joint actions.

**Theorem 1** (Containment of optimal joint actions). *Let $\mathcal{A}_{tgm} := \big\{ \boldsymbol{a} \in \mathcal{A} \mid \boldsymbol{a} = \arg\max_{\boldsymbol{a} \in \mathcal{A}} Q^*(s, \boldsymbol{a}) \big\}$ denote the set of truly optimal joint actions. If $Q_r$ converges to $Q^*$, then $A_{tgm} \subseteq A_r$.*

Thus $A_r$ is guaranteed not to exclude optimal actions, which is critical for policy improvement. Proofs are given in Appendix B.

### 3.3 RECOGNITION-GUIDED WEIGHTING FUNCTION

We now define the POW weighting function:

$$w(s, \boldsymbol{a}) = \begin{cases} 1, & \boldsymbol{a} \in A_r, \\ \alpha, & \boldsymbol{a} \notin A_r, \ \alpha \in [0, 1), \end{cases} \tag{9}$$

where $\alpha$ down-weights actions outside $A_r$. In all our experiments, we set $\alpha = 0$, so only actions in $A_r$ contribute to updates, aligning theory with practice. The training objective for the factorized value network $Q_{tot}$ is then:

$$\mathcal{L}_{Q_{tot}} = \mathbb{E}\big[ w(s, \boldsymbol{a})(Q_{tot}(s, \boldsymbol{a}) - y)^2 \big], \tag{10}$$

with target

$$y = r + \gamma \hat{Q}^*(s', \arg\max_{\boldsymbol{a} \in \mathcal{A}} Q_{tot}(s', \boldsymbol{a})), \tag{11}$$

where $\hat{Q}^*$ is an unrestricted joint value estimator used to approximate $Q^*$.

**Theorem 2** (Convergence of weighted training). *Under the weighting scheme in Eqn. 9, if $A_r$ converges to contain only optimal joint actions, then $Q_{tot}$ recovers the optimal policy.*

If $Q_{tot}$ can recover the joint action with the maximal value of $\hat{Q}^*$, that is, when $\arg\max_{\boldsymbol{a} \in \boldsymbol{A}} Q_{tot}(\boldsymbol{\tau}', \boldsymbol{a}) = \arg\max_{\boldsymbol{a} \in \boldsymbol{A}} \hat{Q}^*(\boldsymbol{\tau}', \boldsymbol{a})$, $\hat{Q}^*$ becomes the optimal joint action value function $Q^*$ according to the Bellman equations indicated by Eqn. 4 and Eqn. 11. Thus $Q_{tot}$ can learn the optimal policy.

Detailed proofs are given in Appendix B.

### 3.4 ITERATIVE WEIGHTED TRAINING

POW proceeds iteratively: (1) update $Q_r$ toward approximating $\hat{Q}^*$ using supervised targets, (2) update $Q_{tot}$ using the weighting $w(s, \boldsymbol{a})$ defined by the current $A_r$, and (3) update $\hat{Q}^*$ based on the updated $Q_{tot}$. This recognition–weighting loop continues throughout training. Unlike heuristic approximations in CW-QMIX or OW-QMIX, this iterative scheme ensures that $A_r$ progressively contracts toward the true optimal set, closing the gap between theoretical guarantees and practical implementation. The pseudocode is provided in Alg. 1.

## 4 EXPERIMENTS

In this section, we instantiate our framework with QMIX and propose the POW-QMIX algorithm. We first evaluate on matrix games and a difficulty-enhanced predator–prey task, both of which exhibit strong non-monotonicity that challenges monotonic value factorization. We then test on the

---

**Algorithm 1** POW Training Procedure

---

**Require:** Replay buffer $\mathcal{D}$; value networks $Q_{\text{tot}}, Q_r, \hat{Q}^*$; tolerance $C$
 1: Initialize all network parameters
 2: **for** each training iteration **do**
 3:  Sample a batch of episodes $\mathcal{B} = \{(\mathbf{o}_{1:T}, \mathbf{a}_{1:T}, r_{1:T})'\}$ from $\mathcal{D}$
 4:  Input observations into agent networks to generate histories:
 5:  $\mathcal{H} = \{(\boldsymbol{\tau}_1, \mathbf{a}_1, r_1, \boldsymbol{\tau}_1'; \ldots; \boldsymbol{\tau}_T, \mathbf{a}_T, r_T, \boldsymbol{\tau}_T')\}$ where $\boldsymbol{\tau}_t' = \boldsymbol{\tau}_{t+1}$
 6:  **for** each time step $t$ and episode in batch **do**
 7:   Compute greedy next action under factorized critic by $\arg\max_{\mathbf{a}} Q_{\text{tot}}(\boldsymbol{\tau}', \mathbf{a})$.
 8:   Compute TD target $y$ shared by all critics by Eqn. 11.
 9:   Update recognition network $Q_r$ by minimizing Eqn. 4c.
10:   Determine greedy action for recognition module under IGM: $\arg\max_{\mathbf{a}} Q_r(\boldsymbol{\tau}, \mathbf{a})$
11:   Compute training weight $w$ based on Eqn. 8 and Eqn. 9.
12:   Update factorized critic $Q_{\text{tot}}$ by Eqn. 10.
13:   Update unconstrained value estimator $\hat{Q}^*$ by Eqn. 4a.
14:  **end for**
15: **end for**
16: **return** $Q_{\text{tot}}$

---

SMAC, a widely used but relatively monotonic benchmark. Finally, we extend the evaluation to SMACv2 and a highway-env intersection scenario.

We also conduct ablations to examine (i) the applicability of POW to other value decomposition methods, and (ii) the effect of increased network size. All experiments are implemented using PyMARL2 (Hu et al., 2021). Hyperparameters such as optimizer type and replay buffer size are tuned for each method. Further details are provided in Appendix F. All results are averaged over five independent runs with different random seeds and are reported with means and 95% confidence intervals.

## 4.1 MATRIX GAME

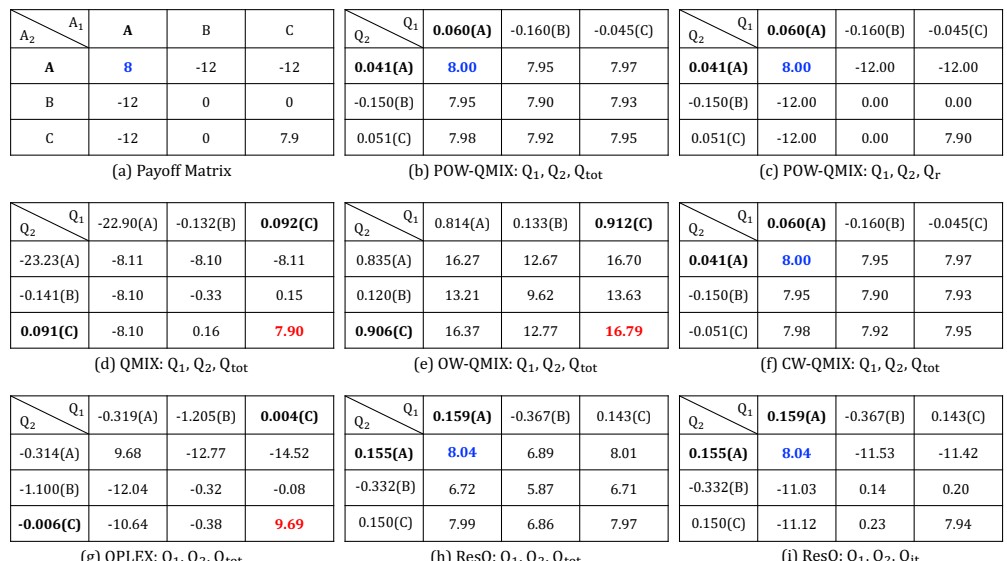

Figure 2: Payoff matrix of a one-step matrix game and reconstructed joint and individual values. Boldface indicates greedy actions. Blue denotes the true optimal joint action, red denotes suboptimal joint actions.

We begin with a coordination matrix game exhibiting strong non-monotonicity, following the setting of ResQ. To remove effects of exploration randomness, we use $\epsilon = 1$ in $\epsilon$-greedy, producing a

uniform data distribution. After convergence, we record the learned joint and individual values (including $Q_r$ for POW-QMIX and $Q_{jt}$ for ResQ), shown in Fig. 2.

POW-QMIX, CW-QMIX, and ResQ successfully recover the optimal policy. In POW-QMIX, the $Q_r$ module precisely estimates the values of all joint actions, enabling accurate recognition of the optimal set $\boldsymbol{A}_r$ for weighting. In contrast, QMIX converges to a locally optimal solution due to monotonicity. OW-QMIX also fails, reflecting its approximation limitations. QPLEX shows partial overestimation of the optimal joint action while remaining accurate elsewhere—an observation that inspired our $Q_r$ design.

## 4.2 PREDATOR–PREY

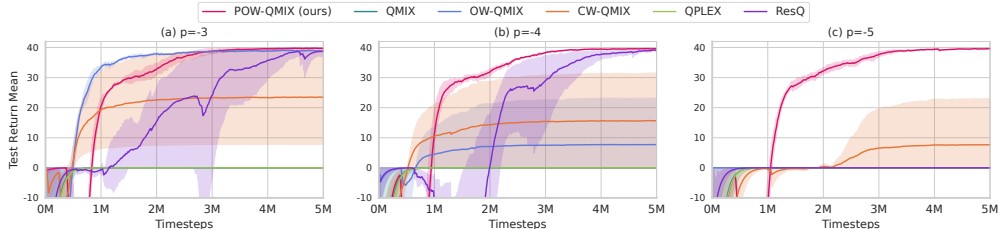

Figure 3: Test return in Predator–Prey with three different mis-capture penalties.

In this task, predators must cooperate to capture prey. If an agent attempts *capture* without coordination, all agents receive a penalty $p$. Higher $|p|$ increases non-monotonicity and encourages passive strategies.

Fig. 3 shows test returns under three penalties. POW-QMIX is the only method that consistently learns the optimal cooperative strategy across all settings. This highlights its ability to resolve non-monotonic structures where baselines fail.

## 4.3 SMAC

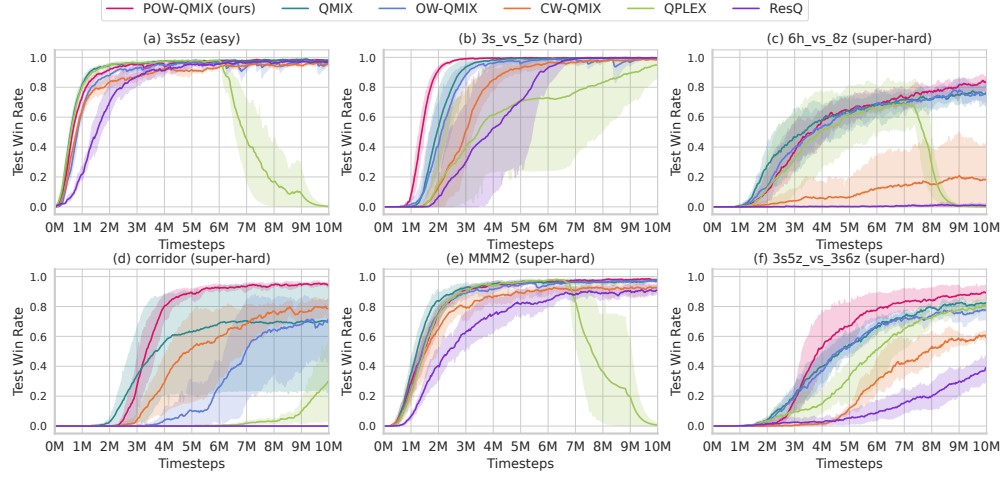

Figure 4: Test win rate on SMAC maps.

We evaluate on six SMAC maps: one easy, one hard, and four super-hard. Fig. 4 shows that POW-QMIX achieves strong performance across maps. Although SMAC is mostly monotonic (Hu et al., 2021), POW-QMIX still matches or outperforms baselines. CW-QMIX, while successful in matrix games, struggles to scale here. QPLEX exhibits instability due to its dueling architecture. OW-QMIX performs well in SMAC but lacks theoretical guarantees, as shown in Fig. 2.

## 4.4 EVALUATION ON HIGHWAY-ENV INTERSECTION AND SMACv2

We include experiments on the highway-env intersection task (Leurent, 2018) and on SMACv2 (Ellis et al., 2023). These benchmarks introduce safety-critical decision making and generalization challenges, complementing the main results.

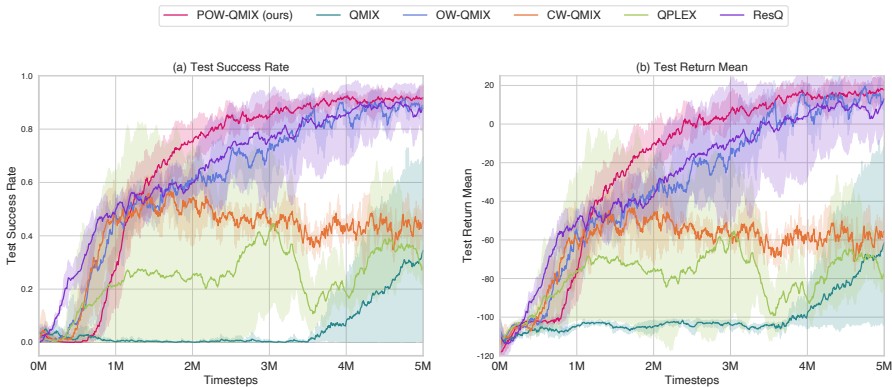

Figure 5: Test return in the highway-env intersection scenario.

As shown in Fig. 5, POW-QMIX achieves the best overall performance, successfully balancing safety and efficiency. In comparison, CW-QMIX converges to overly conservative policies, QPLEX suffers from training instability, and QMIX learns much more slowly. These results highlight POW-QMIX's superior ability to handle strongly non-monotonic environments.

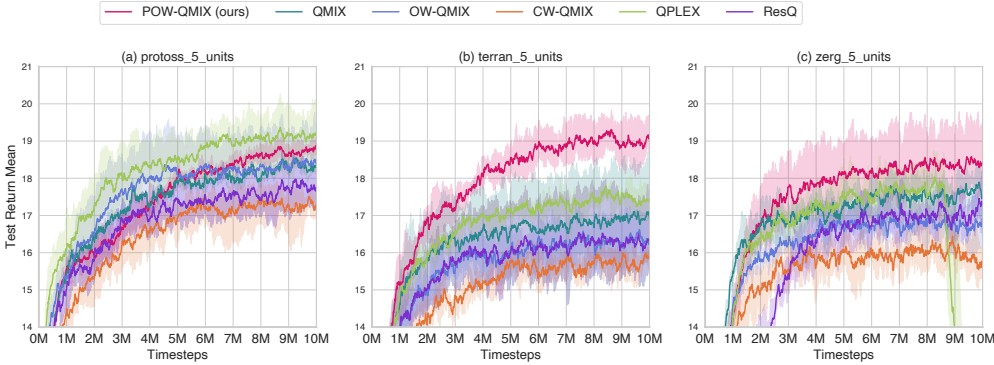

Figure 6: Test return in the SMACv2 benchmarks.

SMACv2 presents more subtle challenges: as win rates of different algorithms often saturate and appear indistinguishable, we adopt average test return as the evaluation metric. Fig. 6 shows that POW-QMIX consistently achieves strong performance across most tasks. While QPLEX outperforms POW-QMIX in the protoss scenario, it collapses in the zerg scenario. Notably, the ablation results in Sec. 4.5.1 show that POW-QPLEX successfully stabilizes QPLEX, confirming that POW's benefits extend beyond QMIX.

## 4.5 ABLATION STUDIES

### 4.5.1 APPLYING POW TO VDN AND QPLEX

We integrate POW into VDN and QPLEX, producing POW-VDN and POW-QPLEX. Tab. 1 summarizes results across all environments, with detailed learning curves in Appendix E. In predator–prey, baseline networks fail to learn, but their POW variants converge quickly to optimal policies. In the highway-env crossroad task, POW greatly improves stability and success rates. On SMAC, POW

Table 1: Performance comparison across environments. Crossroads and SMAC values are win rates; Predator–Prey and SMACv2 values are returns. ↑ indicates improvement over the baseline. Bold indicates best performance.

| Algorithm | Predator-Prey | | Crossroads | SMAC | | | SMACv2 | | |
|---|---|---|---|---|---|---|---|---|---|
| | $p = -4$ | $p = -5$ | | 3s_vs_5z | corridor | MMM2 | protoss | terran | zerg |
| QMIX | 0 | 0 | 0.28 | **1.00** | 0.69 | **0.98** | 18.3 | 17.1 | 17.6 |
| VDN | 0 | 0 | 0.73 | 0.97 | 0.87 | 0.81 | 17.5 | 17.0 | 15.5 |
| QPLEX | 0 | 0 | 0.26 | 0.96 | 0.30 | 0.00 | 19.2 | 17.3 | 0 |
| ResQ | **40** | 0 | 0.88 | **1.00** | 0.00 | 0.90 | 17.7 | 16.3 | 17.5 |
| CW-QMIX | 16 | 8 | 0.43 | 0.99 | 0.79 | 0.92 | 17.2 | 16.0 | 15.7 |
| OW-QMIX | 8 | 0 | 0.88 | **1.00** | 0.70 | **0.98** | 18.4 | 16.3 | 16.9 |
| POW-QMIX | **40** ↑ | **40** ↑ | 0.92 ↑ | **1.00** | **0.95** ↑ | **0.98** | 18.8 ↑ | 19.0 ↑ | **18.4** ↑ |
| POW-VDN | **40** ↑ | **40** ↑ | 0.81 ↑ | 0.96 | 0.87 | 0.90 ↑ | 17.9 ↑ | 17.0 | 16.8 ↑ |
| POW-QPLEX | **40** ↑ | **40** ↑ | **0.93** ↑ | **1.00** ↑ | 0.94 ↑ | 0.93 ↑ | **19.9** ↑ | **19.4** ↑ | 18.1 ↑ |

reduces QPLEX instability, while in SMACv2, all POW variants consistently improve average returns.

### 4.5.2 ENLARGING THE NETWORK SIZE

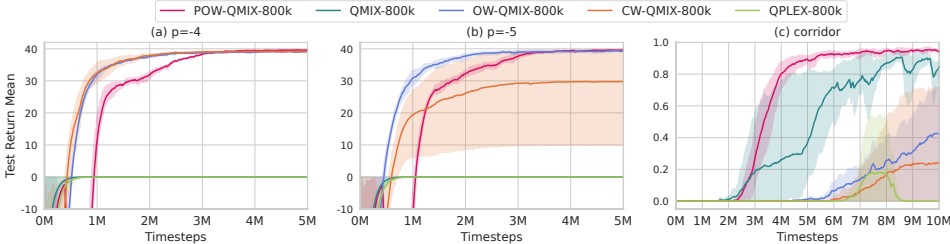

Figure 7: Ablation: effect of network size. (a,b) Predator–Prey with $p = -4, -5$. (c) SMAC map.

To test whether performance gains come simply from added capacity, we enlarge baseline networks to match POW's parameter count. Fig. 7 shows that larger networks improve some baselines (CW-QMIX, OW-QMIX) in predator–prey but hurt in SMAC. Enhanced QMIX improves in SMAC but still fails under non-monotonicity. QPLEX performs poorly regardless of size. Thus, gains of POW stem from its recognition–weighting design, not from parameter count. While $Q_r$ adds moderate complexity (roughly 15–20% training time overhead), POW achieves stronger performance across all tasks. We therefore describe POW as an effective trade-off between computational cost and policy quality.

## 5 RELATED WORK

**Value decomposition in MARL.** Value decomposition is the predominant paradigm under CTDE. VDN (Sunehag et al., 2017) assumes additivity, whereas QMIX (Rashid et al., 2020a) introduces a monotonic mixing network. QPLEX (Wang et al., 2020) improves expressiveness via a dueling structure and advantage-based mixing.

WQMIX (Rashid et al., 2020b) reveals the limitation of uniform weighting and proposes an idealized optimal-action–weighted objective. However, practical variants (CW-QMIX, OW-QMIX) must approximate the optimal action set and therefore cannot guarantee correctness of the assigned weights. Our method is most closely related to WQMIX but differs in two key respects: (1) POW replaces heuristic weighting with a recognition–weighting mechanism that provably converges toward the optimal joint action set; (2) our recognition module $Q_r$ explicitly conditions on the joint action $a$, enabling reliable discrimination among actions, unlike QPLEX where joint-action inputs primarily serve to enhance representational capacity. Thus, POW resolves the gap between theoreti-

cal guarantees and practical realizations that neither WQMIX nor QPLEX addresses. More detailed comparisons are provided in Appendix A.

Beyond these classic methods, CIA (Liu et al., 2023) introduces contrastive identity-aware representation learning to improve credit assignment, and VDT (Zhao et al., 2025) leverages transformers to exploit temporal structure in multi-agent trajectories. Although effective, these methods are orthogonal to our focus: they enhance representation quality or temporal modeling rather than addressing the theoretical–practical mismatch in weighted value decomposition. Therefore, they do not directly evaluate the specific problem POW aims to solve.

Approaches such as REMIX (Mei et al., 2023) and concaveQ (Li et al., 2023) introduce alternative structural assumptions (e.g., concavity or regularization). Our method differs by maintaining generality and instead rethinking how potentially optimal joint actions can be recognized and up-weighted during training.

**Approximation error in value-based MARL.** Several works highlight approximation error as a central challenge. ResQ (Shen et al., 2022) mitigates representational bias by injecting joint-action terms, while CW-QMIX and OW-QMIX approximate weighted learning heuristically and lack optimality guarantees. POW shares the motivation of reducing approximation error but introduces a new mechanism—joint-action conditioning via $Q_r$ and iterative recognition-guided weighting—that ensures optimal actions are retained without requiring exhaustive search.

**Beyond value decomposition.** Actor–critic MARL methods such as MADDPG (Lowe et al., 2017) and MAPPO (Yu et al., 2022) do not rely on value factorization but instead employ joint critics or attention-based critics to stabilize training. These methods differ fundamentally from value decomposition and excel in continuous-action or competitive settings. Our focus is on cooperative discrete-action tasks, where value decomposition remains the most effective and widely used approach. Nevertheless, POW can be viewed as complementary to actor–critic MARL, as both aim to identify joint action structures that improve stability and performance.

## 6 CONCLUSIONS AND LIMITATIONS

We introduced Potentially Optimal Joint Actions Weighting (POW), an iterative weighted training framework for cooperative multi-agent reinforcement learning. POW leverages a recognition module $Q_r$ to identify potentially optimal joint actions and guides training by adaptively weighting them. We formally proved that under this scheme, the recognized set converges to the true optimal joint actions, ensuring that $Q_{tot}$ recovers the optimal policy. Extensive experiments across matrix games, predator–prey, SMAC, SMACv2, and highway-env confirm that POW not only matches its theoretical guarantees but also achieves superior empirical performance over strong baselines.

Despite these advantages, POW introduces additional modules to address non-monotonicity, which increase training complexity in large-scale environments. Moreover, our current study is limited to cooperative settings with discrete action spaces under CTDE. Extending POW to policy-gradient or actor–critic frameworks (e.g., MAPPO) would broaden its applicability to continuous control and mixed cooperative–competitive domains.

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

## A    RELATIONSHIP TO RELATED WORKS

This section clarifies the relationship of POW to prior value decomposition MARL methods. Although POW draws inspiration from both WQMIX and QPLEX, its design addresses their key limitations and provides distinct contributions.

**POW vs. WQMIX.**    WQMIX (Rashid et al., 2020b) proposes an idealized scheme in which optimal joint actions are assigned higher training weights. In principle, this enables recovery of the optimal value function. However, its practical implementations (CW-QMIX, OW-QMIX) must approximate the optimal actions using heuristic strategies, which introduces unavoidable approximation error and prevents strong guarantees. POW differs fundamentally: we introduce a recognition module $Q_r$ that explicitly incorporates joint actions $\boldsymbol{a}$ as inputs and adaptively identifies a set of potentially optimal joint actions $\boldsymbol{A}_r$. The weighting function then prioritizes actions in $\boldsymbol{A}_r$. We provide theoretical analysis (Theorem 1 and 2) showing that $\boldsymbol{A}_r$ converges toward containing only optimal actions, thereby eliminating the approximation gap present in WQMIX. Thus, POW achieves the theoretical guarantee envisioned by WQMIX without resorting to exhaustive search or heuristic approximations.

**POW vs. QPLEX.**    QPLEX (Wang et al., 2020) enhances representational capacity through a dueling-based decomposition that incorporates joint-action–dependent advantage terms. Although this also conditions on joint actions, the primary goal is to increase expressiveness of the value function rather than to guide training dynamics. By contrast, POW leverages joint action inputs within $Q_r$ for a fundamentally different purpose: recognizing potentially optimal joint actions and using them to drive a principled weighting scheme. Ablation results (Tab. 1 and Appendix E) confirm that POW's performance gains cannot be explained merely by including joint action information. Instead, they arise from the recognition–weighting mechanism, which explicitly aligns the training process with the optimal joint value function and provides theoretical guarantees absent in QPLEX.

**POW vs. ResQ.**    ResQ (Shen et al., 2022) introduces an auxiliary joint-action value term to reduce representational bias. However, its objective remains to approximate $Q_{tot}$ without targeted weighting of potentially optimal joint actions. POW differs by explicitly reweighting the learning process toward recognized optimal actions, offering a more direct mechanism to recover optimal policies. As shown in our experiments (Fig. 2, Tab. 1), POW outperforms ResQ in environments with strong non-monotonicity.

## B    PROOF OF THEOREMS

In this section we provide detailed proofs of the main theoretical results. Compared with the original WQMIX analysis, our derivations clarify why the proposed recognition module $Q_r$ avoids approximation errors, and how the recognition–weighting mechanism guarantees recovery of the optimal policy. We explicitly restate all assumptions to avoid ambiguity.

## C    PROOF OF THEOREMS

### C.1    LEMMA 1

For any $\boldsymbol{\tau}$ and joint action $\boldsymbol{a} \notin \boldsymbol{A}_{igm}$, if $Q_r$ has converged, it holds that

$$Q_r(\boldsymbol{\tau}, \boldsymbol{a}) = \min(Q_r(\boldsymbol{\tau}, \hat{\boldsymbol{a}}), Q^*(\boldsymbol{\tau}, \boldsymbol{a})).$$

*Proof.* By the definition of $Q_r$, for any $\boldsymbol{a} \notin \boldsymbol{A}_{igm}$ we have

$$Q_r(\boldsymbol{\tau}, \boldsymbol{a}) \le Q_r(\boldsymbol{\tau}, \hat{\boldsymbol{a}}).$$

The local loss for each joint action is

$$\mathcal{L}_{Q_r}(\boldsymbol{\tau}, \boldsymbol{a}) = \big(Q_r(\boldsymbol{\tau}, \boldsymbol{a}) - Q^*(\boldsymbol{\tau}, \boldsymbol{a})\big)^2.$$

Consider two cases:

- If $Q^*(\boldsymbol{\tau}, \boldsymbol{a}) \geq Q_r(\boldsymbol{\tau}, \hat{\boldsymbol{a}})$, then

$$(Q_r(\boldsymbol{\tau}, \boldsymbol{a}) - Q^*(\boldsymbol{\tau}, \boldsymbol{a}))^2 \geq (Q_r(\boldsymbol{\tau}, \hat{\boldsymbol{a}}) - Q^*(\boldsymbol{\tau}, \boldsymbol{a}))^2.$$

  Minimizing $\mathcal{L}_{Q_r}(\boldsymbol{\tau}, \boldsymbol{a})$ requires $Q_r(\boldsymbol{\tau}, \boldsymbol{a})$ to be as large as possible under the constraint $Q_r(\boldsymbol{\tau}, \boldsymbol{a}) \leq Q_r(\boldsymbol{\tau}, \hat{\boldsymbol{a}})$, yielding

$$Q_r(\boldsymbol{\tau}, \boldsymbol{a}) = Q_r(\boldsymbol{\tau}, \hat{\boldsymbol{a}}) = \min(Q_r(\boldsymbol{\tau}, \hat{\boldsymbol{a}}), Q^*(\boldsymbol{\tau}, \boldsymbol{a})).$$

- If $Q^*(\boldsymbol{\tau}, \boldsymbol{a}) < Q_r(\boldsymbol{\tau}, \hat{\boldsymbol{a}})$, the loss is minimized when

$$Q_r(\boldsymbol{\tau}, \boldsymbol{a}) = Q^*(\boldsymbol{\tau}, \boldsymbol{a}) = \min(Q_r(\boldsymbol{\tau}, \hat{\boldsymbol{a}}), Q^*(\boldsymbol{\tau}, \boldsymbol{a})).$$

Combining both cases completes the proof.

## C.2 LEMMA 2

Let $Q_r$ have converged. Then it holds that

$$Q_r(\boldsymbol{\tau}, \hat{\boldsymbol{a}}) \leq Q^*(\boldsymbol{\tau}, \boldsymbol{a}^*),$$

where $\boldsymbol{a}^* \in \boldsymbol{A}_{tgm}$ is any truly optimal joint action.

*Proof.* Suppose, for contradiction, that

$$Q_r(\boldsymbol{\tau}, \hat{\boldsymbol{a}}) > Q^*(\boldsymbol{\tau}, \boldsymbol{a}^*).$$

From Lemma 1, for any $\boldsymbol{a} \notin \boldsymbol{A}_{igm}$ we have

$$Q_r(\boldsymbol{\tau}, \boldsymbol{a}) = \min\big(Q_r(\boldsymbol{\tau}, \hat{\boldsymbol{a}}), Q^*(\boldsymbol{\tau}, \boldsymbol{a})\big) = Q^*(\boldsymbol{\tau}, \boldsymbol{a}).$$

Construct a new function $Q_r'$ based on $Q_r$:

$$Q_r'(\boldsymbol{\tau}, \boldsymbol{a}) = \begin{cases} Q^*(\boldsymbol{\tau}, \boldsymbol{a}^*), & \boldsymbol{a} \in \boldsymbol{A}_{igm}, \\ Q_r(\boldsymbol{\tau}, \boldsymbol{a}), & \boldsymbol{a} \notin \boldsymbol{A}_{igm}. \end{cases}$$

The corresponding loss for $Q_r'$ is

$$
\begin{aligned}
\mathcal{L}_{Q_r'} &= \sum_{\boldsymbol{a} \in \boldsymbol{A}_{igm}} \big(Q_r'(\boldsymbol{\tau}, \boldsymbol{a}) - Q^*(\boldsymbol{\tau}, \boldsymbol{a})\big)^2 + \sum_{\boldsymbol{a} \notin \boldsymbol{A}_{igm}} \big(Q_r(\boldsymbol{\tau}, \boldsymbol{a}) - Q^*(\boldsymbol{\tau}, \boldsymbol{a})\big)^2 \\
&= \sum_{\boldsymbol{a} \in \boldsymbol{A}_{igm} \cap \boldsymbol{A}_r} \big(Q_r'(\boldsymbol{\tau}, \boldsymbol{a}) - Q^*(\boldsymbol{\tau}, \boldsymbol{a})\big)^2 + \sum_{\boldsymbol{a} \in \boldsymbol{A}_{igm} \setminus \boldsymbol{A}_r} \big(Q_r(\boldsymbol{\tau}, \boldsymbol{a}) - Q^*(\boldsymbol{\tau}, \boldsymbol{a})\big)^2 \\
&< \sum_{\boldsymbol{a} \in \boldsymbol{A}_{igm} \cap \boldsymbol{A}_r} \big(Q_r(\boldsymbol{\tau}, \hat{\boldsymbol{a}}) - Q^*(\boldsymbol{\tau}, \boldsymbol{a})\big)^2 + \sum_{\boldsymbol{a} \in \boldsymbol{A}_{igm} \setminus \boldsymbol{A}_r} \big(Q_r(\boldsymbol{\tau}, \boldsymbol{a}) - Q^*(\boldsymbol{\tau}, \boldsymbol{a})\big)^2 \\
&= \mathcal{L}_{Q_r}.
\end{aligned}
$$

Since $Q_r$ is assumed to have fully converged, the loss cannot be decreased further. But $\mathcal{L}_{Q_r'} < \mathcal{L}_{Q_r}$ under the assumption $Q_r(\boldsymbol{\tau}, \hat{\boldsymbol{a}}) > Q^*(\boldsymbol{\tau}, \boldsymbol{a}^*)$, which is a contradiction.

Hence, we must have

$$Q_r(\boldsymbol{\tau}, \hat{\boldsymbol{a}}) \leq Q^*(\boldsymbol{\tau}, \boldsymbol{a}^*),$$

and Lemma 2 holds.

## C.3 THEOREM 1 CONTAINMENT OF OPTIMAL JOINT ACTIONS

For any $\boldsymbol{\tau}$ and joint action $\boldsymbol{a}$, let $Q_r$ have converged. Then we have

$$\boldsymbol{A}_{tgm} \subseteq \boldsymbol{A}_r,$$

i.e., all truly optimal joint actions are contained in the potentially optimal set $A_r$.

*Proof.* Consider any $\boldsymbol{a}^* \in \boldsymbol{A}_{tgm}$.

- If $\boldsymbol{a}^* \in \boldsymbol{A}_{igm}$, then by definition $\boldsymbol{A}_{igm} \subseteq \boldsymbol{A}_r$, and thus $\boldsymbol{a}^* \in \boldsymbol{A}_r$.
- If $\boldsymbol{a}^* \notin \boldsymbol{A}_{igm}$, from Lemma 1 we have

$$Q_r(\boldsymbol{\tau}, \boldsymbol{a}^*) = \min(Q_r(\boldsymbol{\tau}, \hat{\boldsymbol{a}}), Q^*(\boldsymbol{\tau}, \boldsymbol{a}^*)) = Q_r(\boldsymbol{\tau}, \hat{\boldsymbol{a}}).$$

By Lemma 2, $Q_r(\boldsymbol{\tau}, \hat{\boldsymbol{a}}) \leq Q^*(\boldsymbol{\tau}, \boldsymbol{a}^*)$, so

$$Q_r(\boldsymbol{\tau}, \boldsymbol{a}^*) \geq Q_r(\boldsymbol{\tau}, \hat{\boldsymbol{a}}) - C,$$

and therefore $\boldsymbol{a}^* \in \boldsymbol{A}_r$.

Since every $\boldsymbol{a}^* \in \boldsymbol{A}_{tgm}$ is included in $A_r$, we conclude that

$$\boldsymbol{A}_{tgm} \subseteq \boldsymbol{A}_r.$$

This completes the proof.

### C.4    LEMMA 3

When $Q_r$ has converged:

- If $\boldsymbol{A}_{igm} \subseteq \boldsymbol{A}_{tgm}$, then

$$Q_r(\boldsymbol{\tau}, \hat{\boldsymbol{a}}) = Q^*(\boldsymbol{\tau}, \boldsymbol{a}^*).$$

- If $\boldsymbol{A}_{igm} \nsubseteq \boldsymbol{A}_{tgm}$, then

$$\min_{\boldsymbol{a} \in \boldsymbol{A}_{igm}} Q^*(\boldsymbol{\tau}, \boldsymbol{a}) < Q_r(\boldsymbol{\tau}, \hat{\boldsymbol{a}}) < Q^*(\boldsymbol{\tau}, \boldsymbol{a}^*).$$

*Proof.*

- If $\boldsymbol{A}_{igm} \subseteq \boldsymbol{A}_{tgm}$, then setting $Q_r(\boldsymbol{\tau}, \hat{\boldsymbol{a}}) = Q^*(\boldsymbol{\tau}, \boldsymbol{a}^*)$ achieves $\mathcal{L}_{Q_r} = 0$. Any other value leads to $\mathcal{L}_{Q_r} > 0$, so the minimum is achieved exactly when $Q_r(\boldsymbol{\tau}, \hat{\boldsymbol{a}}) = Q^*(\boldsymbol{\tau}, \boldsymbol{a}^*)$.
- If $\boldsymbol{A}_{igm} \nsubseteq \boldsymbol{A}_{tgm}$, split the loss $\mathcal{L}_{Q_r}$ into

$$\mathcal{L}_1 = \sum_{\boldsymbol{a} \in \boldsymbol{A}_{igm} \cup \boldsymbol{A}_{tgm}} \left( Q_r(\boldsymbol{\tau}, \boldsymbol{a}) - Q^*(\boldsymbol{\tau}, \boldsymbol{a}) \right)^2,$$

$$\mathcal{L}_2 = \sum_{\boldsymbol{a} \notin \boldsymbol{A}_{igm} \cup \boldsymbol{A}_{tgm}} \left( Q_r(\boldsymbol{\tau}, \boldsymbol{a}) - Q^*(\boldsymbol{\tau}, \boldsymbol{a}) \right)^2.$$

By Lemmas 1 and 2, for $\boldsymbol{a} \in \boldsymbol{A}_{igm} \cup \boldsymbol{A}_{tgm}$, $Q_r(\boldsymbol{\tau}, \boldsymbol{a}) = Q_r(\boldsymbol{\tau}, \hat{\boldsymbol{a}})$, so

$$\mathcal{L}_1 = \sum_{\boldsymbol{a} \in \boldsymbol{A}_{igm} \cup \boldsymbol{A}_{tgm}} \left( Q_r(\boldsymbol{\tau}, \hat{\boldsymbol{a}}) - Q^*(\boldsymbol{\tau}, \boldsymbol{a}) \right)^2.$$

Consider $\mathcal{L}_1$ as a quadratic function of $Q_r(\boldsymbol{\tau}, \hat{\boldsymbol{a}})$. Its minimum $m$ satisfies

$$\min_{\boldsymbol{a} \in \boldsymbol{A}_{igm}} Q^*(\boldsymbol{\tau}, \boldsymbol{a}) < m < Q^*(\boldsymbol{\tau}, \boldsymbol{a}^*).$$

For $\mathcal{L}_2$, since $Q_r(\boldsymbol{\tau}, \boldsymbol{a}) \leq Q_r(\boldsymbol{\tau}, \hat{\boldsymbol{a}})$ by Lemma 1, it is monotonically decreasing for $Q_r(\boldsymbol{\tau}, \hat{\boldsymbol{a}}) < \max_{\boldsymbol{a} \notin \boldsymbol{A}_{igm} \cup \boldsymbol{A}_{tgm}} Q^*(\boldsymbol{\tau}, \boldsymbol{a})$, and constant for $Q_r(\boldsymbol{\tau}, \hat{\boldsymbol{a}}) \geq \max_{\boldsymbol{a} \notin \boldsymbol{A}_{igm} \cup \boldsymbol{A}_{tgm}} Q^*(\boldsymbol{\tau}, \boldsymbol{a})$.

Combining $\mathcal{L}_1$ and $\mathcal{L}_2$, the global minimum of $\mathcal{L}_{Q_r} = \mathcal{L}_1 + \mathcal{L}_2$ occurs at a value

$$\min_{\boldsymbol{a} \in \boldsymbol{A}_{igm}} Q^*(\boldsymbol{\tau}, \boldsymbol{a}) < Q_r(\boldsymbol{\tau}, \hat{\boldsymbol{a}}) < Q^*(\boldsymbol{\tau}, \boldsymbol{a}^*),$$

establishing the second case.

This completes the proof.

## C.5 THEOREM 2 (CONVERGENCE OF WEIGHTED TRAINING)

Assuming $Q_{tot}$ satisfies IGM and has a unique maximal joint action $\hat{a}$, there exists $\alpha = 0$ such that $Q_{tot}$ converges with $\hat{a} \in A_{tgm}$ and $A_r = A_{tgm}$.

*Proof.*

We consider the weighted loss for $Q_{tot}$:

$$\mathcal{L}_{Q_{tot}} = \sum_{a} w(s, a) \big( Q_{tot}(\tau, a) - Q^*(\tau, a) \big)^2.$$

Partition joint actions as in the method section:

- $a = \hat{a}$,
- $a \in A_r, a \neq \hat{a}, Q^*(\tau, a) \geq Q_{tot}(\tau, \hat{a})$,
- $a \in A_r, a \neq \hat{a}, Q^*(\tau, a) < Q_{tot}(\tau, \hat{a})$,
- $a \notin A_r$ (weighted by $\alpha$).

When $\alpha = 0$, the last term is zero. Then we exclude the third term and get a lower bound:

$$\mathcal{L}_{Q_{tot}} \geq (Q_{tot}(\tau, \hat{a}) - Q^*(\tau, \hat{a}))^2 + \sum_{\substack{a \in A_r, a \neq \hat{a} \\ Q^*(\tau, a) \geq Q_{tot}(\tau, \hat{a})}} \big( Q_{tot}(\tau, a) - Q^*(\tau, a) \big)^2.$$

Similarly, after $Q_r$ converges, its loss takes the same form:

$$\mathcal{L}_{Q_r} = (Q_r(\tau, \hat{a}) - Q^*(\tau, \hat{a}))^2 + \sum_{\substack{a \in A_r, a \neq \hat{a} \\ Q^*(\tau, a) \geq Q_r(\tau, \hat{a})}} \big( Q_r(\tau, a) - Q^*(\tau, a) \big)^2.$$

Define $Q_r(\tau, \hat{a}) = m$ at the minimum of $\mathcal{L}_{Q_r}$. And for joint actions that satisfy $a \in A_r, a \neq \hat{a}, Q^*(\tau, a) \geq Q_{tot}(\tau, \hat{a}), Q_{tot}(\tau, a) \leq Q_{tot}(\tau, \hat{a})$, $Q_{tot}(\tau, a)$ should be as large as possible and finaly equal to $Q_{tot}(\tau, a)$. Therefore, the minimum values of $\mathcal{L}_{Q_{tot}}$ and $\mathcal{L}_{Q_r}$ are actually the same. We can then construct a valid $Q_{tot}$ satisfying all consumptions:

$$Q_{tot}(\tau, a) = \begin{cases} m + \epsilon, & a = \hat{a}, \\ m, & a \neq \hat{a}, \end{cases}$$

where $\epsilon$ ensures a unique maximal joint action.

Two cases arise:

- If $\hat{a} \in A_{tgm}$, then $Q_{tot}(\tau, \hat{a}) = Q_r(\tau, \hat{a}) = Q^*(\tau, a^*)$, and $Q_{tot}$ has converged.
- If $\hat{a} \notin A_{tgm}$, Lemma 3 gives $Q^*(\tau, \hat{a}) < m < Q^*(\tau, a^*)$.
  Construct

  $$Q'_{tot}(\tau, a) = \begin{cases} Q^*(\tau, a^*), & a = a^*, \\ m, & a \neq a^*, \end{cases}$$

  which satisfies $\mathcal{L}_{Q'_{tot}} < \mathcal{L}_{Q_{tot}}$, ensuring iterative training that moves $\hat{a}$ toward $A_{tgm}$.

Thus, with iterative training and $\alpha = 0$, $Q_{tot}$ converges such that $\hat{a} \in A_{tgm}$ and $A_r = A_{tgm}$. The result also holds for $Q_{tot}$ satisfying IGM without strict monotonicity, e.g., QPLEX.

## C.6 REMARK ON THE UNIQUE MAXIMAL JOINT ACTION ASSUMPTION

In Theorem 2, we assume that $Q_{tot}$ has a unique maximal joint action $\hat{a}$ for simplicity of analysis. In practice, this assumption can be relaxed:

- Even if multiple joint actions achieve the same maximal value, the weighted training procedure will assign higher emphasis to those in $A_{tgm}$, guiding the learning dynamics toward the set of potentially optimal joint actions.

- The uniqueness can also be enforced with an arbitrarily small perturbation $\epsilon$ added to break ties, which does not affect policy performance but ensures theoretical convergence of the proof.
- Empirically, in stochastic environments or with function approximation, exact ties are rare, so this assumption is reasonable for most practical multi-agent RL tasks.

Thus, the assumption mainly simplifies the theoretical exposition without restricting the practical applicability of the method.

## D DISCUSSION

This section provides an accessible discussion of the motivation and design rationale behind POW, clarifying the innovations and avoiding potential confusion with existing value factorization methods.

### D.1 CORE INNOVATIONS

The novelty of POW is reflected in two key aspects: (1) It eliminates the need to traverse the entire exponentially large joint action space when recognizing optimal joint actions; (2) It provides a theoretical guarantee of convergence to the global optimum, without introducing approximation error in practice.

These advantages directly address the limitations of prior approaches such as WQMIX and QPLEX.

### D.2 PROBLEM CONTEXT

Within the CTDE framework, the IGM condition requires training on a centralized $Q_{tot}$ function. Due to the monotonicity constraints imposed by mixing networks (e.g., QMIX), a joint action may be incorrectly undervalued when some agents take suboptimal actions. This prevents accurate estimation of globally optimal joint actions.

WQMIX mitigates this by reweighting potentially optimal joint actions more heavily during training (Rashid et al., 2020b). However, identifying these actions requires an unrestricted value function over the full joint action space. Since the joint action space grows exponentially with the number of agents, WQMIX resorts to approximations that inevitably introduce error, limiting its practical applicability.

### D.3 DESIGN RATIONALE OF $Q_r$

POW avoids the drawbacks of WQMIX by introducing a recognition module, $Q_r$, that directly identifies a superset of potentially optimal joint actions, denoted $\boldsymbol{A}_r$. Instead of exhaustively searching over all joint actions, POW uses $\hat{\boldsymbol{a}} = \arg\max_{\boldsymbol{a}} Q_r(\boldsymbol{\tau}, \boldsymbol{a})$ as a reference and recognizes $\boldsymbol{A}_r$ without approximation. This set is then weighted more strongly during training of $Q_{tot}$, ensuring accurate estimation of globally optimal policies.

To achieve this, $Q_r$ is designed with three essential properties:

**1. Independence of joint action values.** $Q_r$ explicitly takes the joint action $\boldsymbol{a}$ as input, ensuring that $Q_r(\boldsymbol{\tau}, \boldsymbol{a})$ is independently parameterized for each action. This avoids the monotonic coupling between joint actions present in QMIX's mixing structure, enabling $Q_r$ to recover the true $Q^*(\boldsymbol{\tau}, \boldsymbol{a})$ values without interference.

**2. Satisfaction of IGM.** Although free from monotonicity, $Q_r$ still satisfies the IGM condition, i.e., $\arg\max_{\boldsymbol{a}} Q_r(\boldsymbol{\tau}, \boldsymbol{a}) = \hat{\boldsymbol{a}}$. This allows $\hat{\boldsymbol{a}}$ to serve as a baseline for identifying all potentially optimal joint actions in $\boldsymbol{A}_r$.

**3. Accurate recovery of $Q^*$.** $Q_r$ is trained against the true joint action values $Q^*$, rather than surrogate targets. Thanks to its independence property, $Q_r$ can precisely match $Q^*(\boldsymbol{\tau}, \boldsymbol{a})$ for each action, ensuring that $\boldsymbol{A}_r$ can be recognized by simple comparison with $Q_r(\boldsymbol{\tau}, \hat{\boldsymbol{a}})$.

## D.4  HOW $Q_r$ ENABLES POW

These three properties guarantee that $Q_r$ recovers the set $\boldsymbol{A}_r$ without approximation, and that $\boldsymbol{A}_r$ gradually contracts to $\boldsymbol{A}_{tgm}$ as training proceeds. This mechanism enables POW to retain the strengths of WQMIX (emphasizing potentially optimal joint actions) while avoiding its reliance on approximations. Unlike QPLEX, which assigns equal weight to all joint actions and often suffers from instability, POW selectively emphasizes $\boldsymbol{A}_r$, ensuring both stability and convergence guarantees.

Fig. 8 provides an intuitive visualization: $Q_r$ establishes a baseline plane at $Q_r(\boldsymbol{\tau}, \hat{\boldsymbol{a}})$, above which potentially optimal actions are recognized. As training proceeds, this plane rises until it aligns with the true global optimum, at which point $\boldsymbol{A}_r = \boldsymbol{A}_{tgm}$ and the optimal policy is recovered.

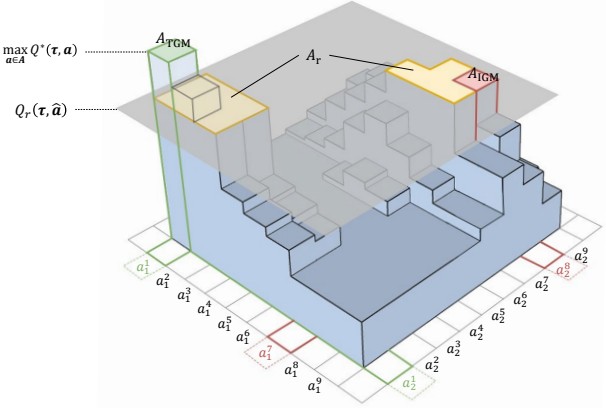

Figure 8: This figure illustrates the $Q^*$-value landscape, where the height of each column represents the $Q^*$-value associated with a particular joint action. (The exact heights are not critical for the concepts discussed herein.) The current convergence state of the $Q_r$ network resembles Stage 2 in Fig. 9. The red area represents $\hat{\boldsymbol{a}}$. The yellow area highlights $A_r$, which is determined via $\hat{\boldsymbol{a}}$ and represents the subset of actions on which POW focuses its weighted training efforts. The green area denotes the global optimal joint actions. For $Q_r$, the Q-values beneath the conceptual plane are already learned, while the Q-values within $A_r$ are set at the plane's level. As the training progresses, the plane is expected to rise incrementally, identifying increasingly higher $Q^*$-values.

## D.5  ILLUSTRATIVE EXAMPLE

**Stage I** → **Stage II** → **Stage III**

$Q_{tot}$

| $A_1 \backslash A_2$ | A | B | C |
|---|---|---|---|
| A | 2.78 | 2.81 | 3.11 |
| B | 2.78 | 2.81 | 3.11 |
| C | 2.78 | 2.91 | **3.21** |

| $A_1 \backslash A_2$ | A | B | C |
|---|---|---|---|
| A | 3.97 | 3.88 | 4.46 |
| B | 3.93 | 3.83 | 4.42 |
| C | 4.13 | 4.03 | **4.62** |

| $A_1 \backslash A_2$ | A | B | C |
|---|---|---|---|
| A | **8.41** | 6.89 | 7.18 |
| B | 7.70 | 6.19 | 6.48 |
| C | 7.87 | 6.36 | 6.65 |

$Q_r$

| $A_1 \backslash A_2$ | A | B | C |
|---|---|---|---|
| A | 3.63 | −12.34 | −12.63 |
| B | −11.8 | −0.11 | −0.03 |
| C | −11.94 | 0.12 | **3.65** |

| $A_1 \backslash A_2$ | A | B | C |
|---|---|---|---|
| A | 7.90 | −12.24 | −11.95 |
| B | −12.06 | 0.05 | 0.04 |
| C | −11.99 | 0.12 | **7.90** |

| $A_1 \backslash A_2$ | A | B | C |
|---|---|---|---|
| A | **8.00** | −12.12 | −12.06 |
| B | −12.06 | −0.03 | −0.05 |
| C | −11.96 | −0.02 | 7.90 |

$Q_i$

| $Q_i$ | A | B | C |
|---|---|---|---|
| $Q_1$ | −0.16 | −0.20 | **0.31** |
| $Q_2$ | −0.17 | −0.18 | **0.31** |

| $Q_i$ | A | B | C |
|---|---|---|---|
| $Q_1$ | −0.06 | −0.21 | **0.43** |
| $Q_2$ | −0.07 | −0.19 | **0.44** |

| $Q_i$ | A | B | C |
|---|---|---|---|
| $Q_1$ | **0.57** | −0.09 | 0.18 |
| $Q_2$ | **0.68** | −0.05 | 0.16 |

Figure 9: Three stages of the matrix game. The potentially optimal joint action is highlighted with a yellow border.

To provide intuition for how POW-QMIX overcomes non-monotonicity, we illustrate its behavior in a one-step matrix game. The joint action space is $\{A, B, C\}$. In such settings, $Q^*$ and $\hat{Q}^*$ are equivalent to the ground-truth reward function, allowing us to directly track the evolution of joint action values during training.

The training process can be divided into three stages (I–III) as depicted in Fig. 9.

**Stage I–II.** At the beginning of training, based on the values estimated by the $Q_r$ module, we can identify $(A, A)$ and $(C, C)$ as potentially optimal joint actions, with weights set to 1, while all other joint actions receive zero weight. During Stage II, the $Q_{tot}$ value for $(C, C)$ already matches $Q^*$, so its gradient vanishes. In contrast, for $(A, A)$, $Q_{tot} < Q^*$, meaning the gradient update increases $Q_{tot}$ and propagates improvements to the corresponding individual utilities $Q_1(\tau_1, A)$ and $Q_2(\tau_2, A)$.

**Stage III.** As training proceeds, $(A, A)$ becomes the only remaining potentially optimal joint action. This action coincides with the true global optimum, enabling POW-QMIX to escape the local optimum (with value 7.9) and converge to the correct solution.

## E  ADDITIONAL RESULTS OF ABLATION STUDIES

We test the generality of POW by applying it to two other value decomposition baselines, yielding POW-VDN and POW-QPLEX. These experiments demonstrate that POW is not tied to a specific base algorithm but provides a general mechanism for improving non-monotonicity handling.

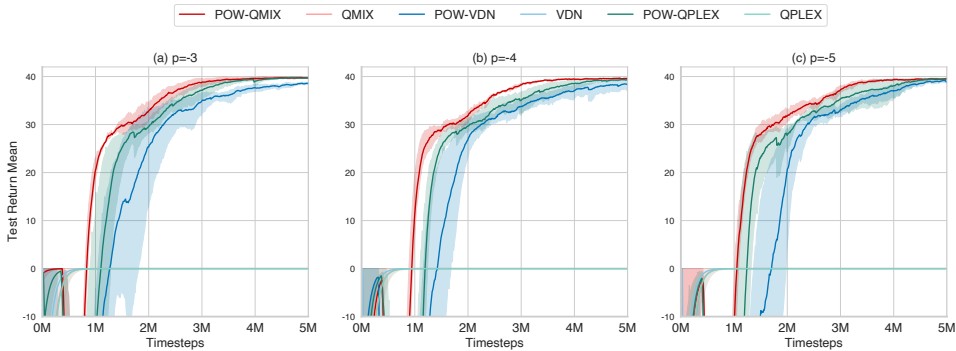

Figure 10: Application of POW to Predator-Prey with three levels of mis-capture penalty.

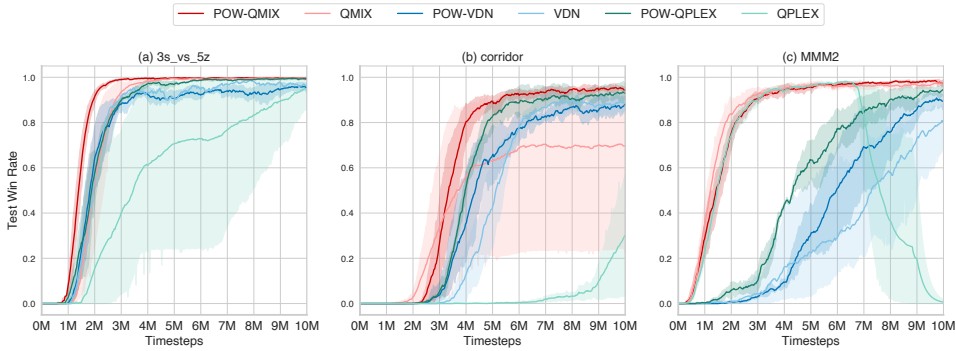

Figure 11: Application of POW to SMAC benchmarks.

The results across Predator-Prey (Fig. 10), SMAC (Fig. 11), highway-env intersection (Fig. 12), and SMACv2 (Fig. 13) consistently show that adding POW substantially improves performance and stability. Importantly, POW-QPLEX alleviates the instability issues commonly observed in QPLEX, and POW-VDN provides noticeable gains despite VDN's limited expressiveness. These findings support the general applicability of POW as a plug-in improvement to value decomposition methods.

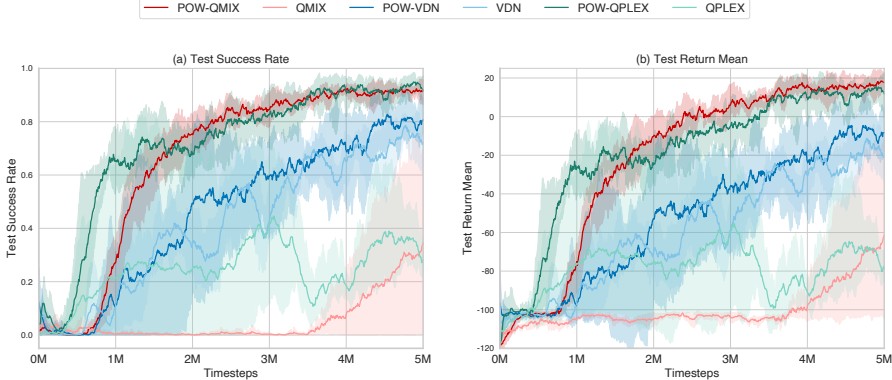

Figure 12: Application of POW to the highway-env intersection scenario.

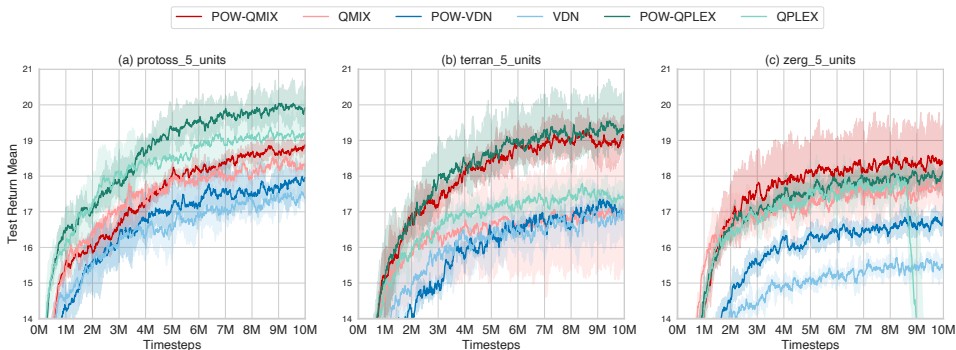

Figure 13: Application of POW to SMACv2 benchmarks.

## F EXPERIMENTAL SETUP

**Note:** In Sec. 4, we set the weight for potentially optimal joint actions to 1 and for all other joint actions to $\alpha \in [0, 1)$, following the weighting function in Equation (10) of the WQMIX paper, which is commonly used in the WQMIX methodology.

We emphasize that Theorem 2 holds strictly when $\alpha = 0$. As stated after the definition, both our theoretical analysis and experimental implementation consistently adopt $\alpha = 0$ to ensure alignment between theory and practice. In practice, setting $\alpha = 0$ avoids introducing approximation errors from down-weighting suboptimal actions, ensuring that only the recognized potentially optimal set contributes to training.

### F.1 PARAMETER SETTINGS FOR BASELINE ALGORITHMS

We conducted all experiments using the PyMARL2 framework, an enhanced version of the original PyMARL, specifically optimized for the StarCraft Multi-Agent Challenge (SMAC). PyMARL2 incorporates several implementation refinements and hyperparameter adjustments to improve performance across various scenarios. There are many code-level tricks in PyMARL2, such as the use of the Adam optimizer, the batch size, the replay buffer size, the rollout processes, the $\epsilon$-greedy exploration strategy, and the $TD(\lambda)$ parameter. These hyperparameters are set to the same values across all algorithms, including ours, to ensure a fair comparison.

The hyperparameters are listed in Tab. 2, Tab. 3, Tab. 4, Tab. 5, and Tab. 6.

Table 2: Common Hyperparameters in Pymarl2

| HYPERPARAMETER | VALUE |
|---|---|
| TRAINING MODE | PARALLEL |
| ROLLOUT PROCESSES | 8 |
| REPLAY BUFFER SIZE | 5000 |
| BATCH SIZE (TRAINING) | 128 |
| ACTION SELECTION | $\epsilon$-GREEDY |
| $\epsilon$ START | 1.0 |
| $\epsilon$ FINISH | 0.05 |
| $\epsilon$ ANNEAL STEPS | 500K |
| OPTIMIZER | ADAM |
| LEARNING RATE | 0.001 |
| TARGET NETWORK UPDATE INTERVAL | 200 |
| $TD(\lambda)$ | 0.6 |
| LAYER NORMALIZATION | FALSE |
| ORTHOGONAL INITIALIZATION | FALSE |
| ORTHOGONAL GAIN | 0.01 |
| PRIORITY EXPERIENCE REPLAY (PER) | FALSE |
| PER $\alpha$ | 0.6 |
| PER $\beta$ | 0.4 |
| RETURN-BASED PRIORITY | FALSE |
| MIXING EMBEDDING DIMENSION | 32 |
| HYPERNETWORK EMBEDDING | 64 |
| HYPERNETWORK LAYERS | 2 |

Table 3: QMIX-Specific Hyperparameters

| HYPERPARAMETER | VALUE |
|---|---|
| AGENT ARCHITECTURE | RNN |
| QMIX LOSS WEIGHT | 1.0 |

Table 4: W-QMIX-Specific Hyperparameters

| HYPERPARAMETER | VALUE |
|---|---|
| WEIGHTS FOR OPTIMAL JOINT ACTIONS | 1 |
| WEIGHTS FOR OTHER JOINT ACTIONS | 0.1 |

Table 5: QPLEX-Specific Hyperparameters

| HYPERPARAMETER | VALUE |
|---|---|
| DOUBLE Q-LEARNING | TRUE |
| ADVANTAGE HYPERNETWORK LAYERS | 2 |
| ADVANTAGE HYPERNETWORK EMBEDDING | 64 |
| NUMBER OF KERNELS | 4 |
| MINUS-ONE TRANSFORMATION | TRUE |
| WEIGHTED HEAD | TRUE |
| ADVANTAGE ATTENTION | TRUE |
| GRADIENT STOP MECHANISM | TRUE |

## F.2 MATRIX GAME

In a matrix game environment, two agents independently select actions, forming a joint action to receive an immediate reward. This reward directly reflects the true value of the joint action. This type of environment is characterized by a simple and unique state space, eliminating the need to consider complex state transitions. Simultaneously, the reward is directly equivalent to the true value, requiring no additional modeling. Furthermore, the reward structure can be flexibly designed,

Table 6: ResQ-Specific Hyperparameters

| HYPERPARAMETER | VALUE |
|---|---|
| LEARNER | RESQ CENTRAL LEARNER |
| DOUBLE Q-LEARNING | TRUE |
| MIXING NETWORK | QMIX |
| HYSTERETIC QMIX (CW/OW-QMIX) | FALSE |
| CENTRAL MIXING EMBEDDING | 128 |
| CENTRAL ACTION EMBEDDING | 1 |
| CENTRAL MAC | BASIC CENTRAL MAC |
| CENTRAL AGENT | CENTRAL RNN |
| CENTRAL RNN HIDDEN DIMENSION | 64 |
| CENTRAL MIXER | FEEDFORWARD |
| RESQ VERSION | V3 |
| CENTRAL LOSS WEIGHT | 1.0 |
| NO-OPT LOSS WEIGHT | 1.0 |
| QMIX LOSS WEIGHT | 1.0 |
| CONSTRAINT LOSS TYPE | MSE |
| CONSTRAINT LOSS DELTA | 0.001 |
| MAX SECOND GAP | 0 |
| CONSTRAINT METHOD | MAX ACTION |
| RESIDUAL Q-VALUE ABSOLUTE | TRUE |

facilitating the construction of test scenarios with different characteristics. Lastly, the results are intuitive and easy to analyze and visualize. It is precisely because of these characteristics that matrix games have become an ideal testbed for studying the theoretical performance of value decomposition algorithms.

We set $\epsilon = 1$ throughout the experiments on matrix game to achieve uniform data distribution and set ideal weights for the purpose of theoretical analysis. The weights for potentially optimal joint actions and other joint actions in POW-QMIX are 1 and 0. The weights for optimal joint actions and other joint actions in CW-QMIX and OW-QMIX are 1 and 0. The constant $C$ used in Eqn. 9 is set to 0.05.

## F.3 PREDATOR-PREY

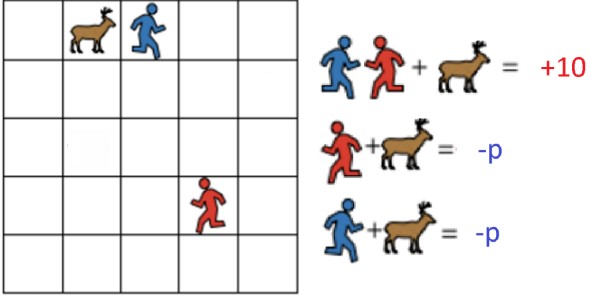

Figure 14: Stag Hunt Game

The "Stag Hunt" game in game theory is a classic scenario that profoundly reveals the inherent conflict between individual rationality and collective rationality, as well as potential coordination mechanisms, while also highlighting the crucial role of trust in fostering cooperation. This tension between individual and collective rationality precisely constitutes the core of the non-monotonicity problem explored in this paper. In the Stag Hunt scenario, agents face two strategic choices: one is a high-risk cooperative strategy, which yields the highest payoff when all participants choose this strategy, but if only a single agent attempts to cooperate while other agents choose not to, that agent will suffer severe losses or even penalties; the other is a low-risk safe strategy, where an agent

adopting this strategy can obtain a stable but relatively low payoff, regardless of the choices of other agents.

Table 7: Predator-Prey Experiment Payoff Matrix

| $A_1 \setminus A_2$ | Move Up | Move Down | Move Left | Move Right | Stay Still | Capture |
|---|---|---|---|---|---|---|
| Move Up | 0 | 0 | 0 | 0 | 0 | $-p$ |
| Move Down | 0 | 0 | 0 | 0 | 0 | $-p$ |
| Move Left | 0 | 0 | 0 | 0 | 0 | $-p$ |
| Move Right | 0 | 0 | 0 | 0 | 0 | $-p$ |
| Stay Still | 0 | 0 | 0 | 0 | 0 | $-p$ |
| Capture | $-p$ | $-p$ | $-p$ | $-p$ | $-p$ | 10 |

The predator-prey environment adopted in this paper is an extension of the Stag Hunt concept within a complex Markov Decision Process. This environment retains the core characteristics of the Stag Hunt game while introducing a richer strategy space and dynamic interactions. In this environment, multiple agents acting as predators need to effectively cooperate to successfully capture the prey. All units (including agents and prey) move and interact in a discrete grid world.

The detailed settings of this environment are as follows: We construct a $10 \times 10$ grid world as the state space, where each grid cell can contain: empty space, an agent, or the prey. Considering the limitations of real-world perception, we limit the observation range of an agent to a $3 \times 3$ grid area centered on itself, allowing it to only perceive the types of units within this range, thus forming a Partially Observable Markov Decision Process (POMDP). The action space of an agent includes six discrete choices: moving in the four cardinal directions (up, down, left, right), staying in place, and performing a capture action.

The reward mechanism is designed to reflect the necessity of cooperation: Only when at least two agents simultaneously perform a capture action in positions adjacent to the prey can the capture be successful, whereupon all agents receive a positive reward of $+10$. Conversely, if only one agent attempts to perform a capture action in isolation, not only will the capture action fail, but that agent will also incur a penalty of $-p$. This design directly maps to the risk-reward trade-off in the Stag Hunt game.

As the absolute value of the mis-capture penalty parameter $p$ increases, the non-monotonic characteristics of the environment become more prominent. A stricter penalty mechanism reinforces the non-monotonicity of the reward structure, prompting agents to be more inclined to adopt conservative strategies—completely avoiding the risk of performing a capture action—thereby potentially missing out on high-payoff cooperative opportunities. This phenomenon provides an ideal test scenario for our research on how algorithms can overcome non-monotonicity limitations.

The default experimental settings are consistent with those in the PyMARL2 framework. The constant $C$ used in Eqn. 8 is set to 1.

### F.4 SMAC

In the PyMARL2 framework, certain parameters such as hidden size and $TD(\lambda)$ have been specifically fine-tuned for the 6h_vs_8z and 3s5z_vs_3s6z maps. However, for the sake of a fair comparison, we set all algorithms to use default parameters across all maps. The constant $C$ used in Eqn. 8 is set to 0.05.

### F.5 SMACv2

The default experimental settings are consistent with those in the PyMARL3 framework. The constant $C$ used in Eqn. 8 is set to 0.05.

### F.6 INTERSECTION SCENARIO IN HIGHWAY-ENV

Highway-env Leurent (2018) is a collection of environments specifically designed for autonomous driving decision-making tasks. Its intersection scenario simulates a complex traffic environment, an

example of which is shown in Fig. 15, providing an ideal platform for us to evaluate the performance of algorithms on non-monotonicity problems.

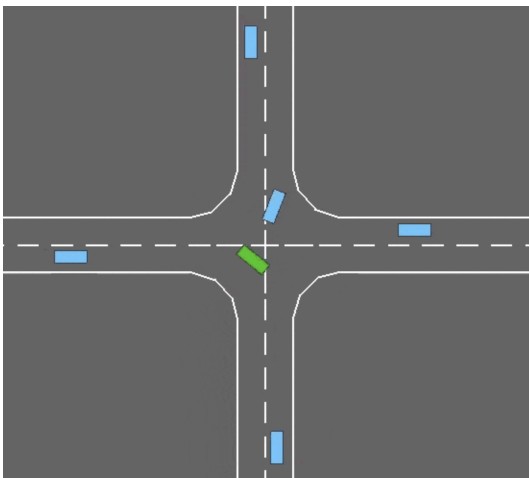

Figure 15: Example of the intersection scenario environment.

In this scenario, multiple vehicles approach an unsignalized intersection from different directions, with each vehicle controlled by an independent agent policy. These vehicles follow pre-planned routes, and the primary task of the agents is to control their vehicle's speed to ensure safe and efficient passage through the intersection. The reward mechanism is intricately designed: a positive reward is given only when all vehicles safely pass through the intersection and reach their respective destinations; conversely, if any collision occurs, all agents not only receive a severe negative penalty, but the current episode also terminates immediately.

This design leads to the environment exhibiting strong non-monotonic characteristics. Due to the significant penalty associated with collisions, agents can easily learn extremely conservative strategies—such as stopping completely and waiting outside the intersection to avoid any potential collision risk. However, while such conservative strategies can avoid penalties, they fail to achieve the positive reward for successfully navigating the intersection, leading to poor overall performance. Therefore, agents need to learn to find a balance between safety and efficiency, making this an ideal scenario for testing an algorithm's ability to handle non-monotonic challenges.

We adopted the same scenario and reward settings as in Huang et al. (2023). The $\epsilon$ value is set to 0.1 to ensure the same data distribution for all algorithms. The constant $C$ used in Eqn. 8 is set to 0.1.

THE USE OF LLMS

We thank ChatGPT-5 for its assistance in polishing the writing and proofreading of this paper. The authors are responsible for the content and presentation.

