# OpenReview forum: "Potentially Optimal Joint Actions Recognition for Cooperative Multi-Agent Reinforcement Learning"
_ICLR.cc/2026/Conference — ICLR 2026 Poster_

### Official Review · Reviewer_LRKB · 2025-10-23

**Soundness:** 2
**Presentation:** 3
**Contribution:** 2
**Rating:** 4
**Confidence:** 4

**Summary:**

This paper proposes a new way to identify the optimal joint actions to weight the learning of value-decomposition methods. An additional recognition $Q$-function is learned, which induces a set of actions that may potentially be optimal. While its learning proceeds toward approximating the optimal values, this set gets further refined, until only the optimal actions are present. Experimental results on a wide set of different problems show the better performances achieved by this method, as well as how it can be combined with general value-decomposition algorithms and improve over their vanilla versions.

**Strengths:**

The issue of overcoming the representation limitations of existing value-decomposition methods, for example through mean of weighting their learning updates as done by this paper, is a key direction to deliver better MARL algorithms. The idea of doing so by identifying the optimal joint actions in an effective way is a viable path for this, and already proved capable of achieving improvements in driving existing methods, as with WQMIX. The proposed set of experiments is very wide and varied, and the reported empirical results are strong. The paper is in general quite clear and easy to follow.

**Weaknesses:**

I struggle to understand some of the implementation choices made here: for example, the use of a separate $Q_r$ over the already existing $Q^\*$ is not entirely justified to me. Also, it is not completely clear how the proposed method should overcome the limitations identified in the existing CW-QMIX and OW-QMIX algorithms. Finally, the cost (in terms of computational time) of the proposed method seems a bit hidden and not sufficiently and clearly highlighted, making it difficult to actually assess the trade-off one has to make when choosing the proposed method over other ones. Please see the Questions below for a more in-details explanation.

**Questions:**

- You claim that, being trained against the optimal $Q^\*$, the maximizing actions of $Q_r$ are going to be the same as those of $Q^\*$ itself, and thus the set $A_r$ will include them. But the training of both $Q_r$ and $Q^\*$ is done simultaneously, and thus we are not guaranteed that their interplay is accurate before convergence occurred. This would probably lead to similar problems to those you highlighted for CW-QMIX and OW-QMIX (i.e., inaccuracies due to learning) no? Am I missing something here?

- I struggle to understand how the additional $Q_r$ is bringing any benefit over the use of $Q^\*$ directly to identify the optimal joint actions: at convergence, these will both represent the same action-value function, no? And $Q_r$ is indeed trained to chase the optimal unrestricted $Q^\*$. So why adding an additional structure rather than simply restructuring $Q^\*$ itself to be formulated as Equation (3)?

- When computing $w(s,\mathbf{a})$, how do we check if $\mathbf{a}\in A_r$? If we need to explicitly construct the set of recognized optimal actions $A_r$, then this may be a quite expensive operation. Such an aspect should be eventually stated clearly.

- It would be good to explicitly state the loss function for training $Q^\*$ in the paper, as currently it can only be found in Figure 1.

- In Figure 2, why are the values for $Q_1$ and $Q_2$ of ResQ different between sub-figure (h) and (i)?

---

> ### Author Response · Authors · 2025-11-23
>
> ## **Response to Questions**
>
> **1. On the Interplay Between $Q_r$ and $Q^\*$ During Training**
>
> You are right to point out the challenges of iterative training.
> In each iteration, the purpose of $Q_r$ is to identify the set of potentially optimal joint actions corresponding to the current target $\hat{Q}^\*$. The weighted training then ensures that $Q\_{tot}$ can also recover the current optimal joint action corresponding to that same $\hat{Q}^\*$. This ensures that the action selected by $Q\_{tot}$ aligns with the current estimated optimal action (i.e., $\arg\max Q\_{tot} = \arg\max \hat{Q}^\*$). Consequently, the practical training target for $\hat{Q}^\*$, which is $r + \gamma \hat{Q}^\*(s', \arg\max_{\boldsymbol{a}'} Q\_{tot}(s', \boldsymbol{a}'))$, is correctly transformed into the ideal target, $r + \gamma \hat{Q}^\*(s', \arg\max_{\boldsymbol{a}'} \hat{Q}^*(s', \boldsymbol{a}'))$. This structure effectively constructs a valid Bellman Optimality Equation, which guarantees that $\hat{Q}^\*$ will ultimately converge to the true $Q^\*$.
>
> **2. On the Benefit of an Additional $Q_r$ Over Using $Q^*$ Directly**
>
> The true optimal action-value function $Q^\*$ is unknown. WQMIX introduced an unrestricted value function, which we denote as $\hat{Q}^\*$, to approximate $Q^\*$. However, the optimal Bellman equation requires computing $\arg\max_{\boldsymbol{a}} \hat{Q}^\*$, which, for an unrestricted function like $\hat{Q}^\*$, necessitates an exhaustive search over the entire joint action space. To avoid this exponential complexity, WQMIX resorts to approximations, creating the gap between its theory and implementation.
>
> Our $Q_r$ module is introduced precisely to solve this problem. It provides a mechanism to identify the set of potentially optimal actions without iterating through the entire joint action space, thus enabling $Q\_{tot}$ to recover the optimal action for $\hat{Q}^\*$ under the iterative weighted training. This closes the theory-practice gap that exists in WQMIX. Please refer to the general response for more detailed description.
>
> **3. On the Computational Cost of Checking $\boldsymbol{a} \in A_r$**
>
> In fact, we do not need to explicitly construct the full set $A_r$. To check if an action $\boldsymbol{a}$ is a potentially optimal joint action (i.e., if it is in $A_r$), we only need to compare the values of $Q_r(s, \boldsymbol{a})$ and $Q_r(s, \hat{\boldsymbol{a}})$. **Therefore, this operation has linear complexity**, not exponential complexity. We have revised the method sections to make this point clearer in the revision.
>
> **4. On Stating the Loss Function for $Q^\*$**
>
> Certainly. We agree this would improve clarity and have added the explicit loss function for training $Q^\*$ in the main paper in the revised version, please see Section 3.1.
>
> **5. On the Inconsistency in Figure 2 for ResQ**
>
> We apologize for this inconsistency. You are correct; since ResQ is able to converge to the global optimum in this matrix game environment, the values for $Q_1$ and $Q_2$ in sub-figures (h) and (i) differ because we did not strictly unify the time steps at which the statistics for $Q_{tot}$ and $Q_{jt}$ were captured. **We have corrected this in the revision to ensure consistency**.
>
> Thanks again for your valuable comments and questions!  If there are any additional questions, we would be happy to address them.

---

> > ### Comment · Reviewer_LRKB · 2025-11-24
> > **Reply to Authors**
> >
> > I would like to thank the authors, whose rebuttal addressed some of my concerns. However, some aspects are still not clear:
> >
> > **1.** I am not sure how your comment addresses my question: having $\hat{Q}^\*$ (I wrote $Q^\*$, I apologize!) to be learned together with $Q_r$ may still lead to situation where the optimal joint actions of $\hat{Q}^\*$ are wrong, pushing $Q_r$ toward an incorrect identification and $Q_{tot}$ toward an incorrect learning in turn. This to me does not sound too different from the issue you highlighted in WQMIX, where basically we have to turn to an heuristic to approximate the defined optimal recognition mechanism (in your case, trusting $\hat{Q}^\*$ to be able to get the optimal joint actions while learning). Could you explain a bit more in details why, using your $Q_r$, we do not have the risk to stumble on wrong optimal joint actions?
> >
> > **2.** I am unsure how this actually works: $Q_r$ and $\hat{Q}^\*$ are learned against the same objective, and are both unrestricted $Q$-functions no? Where does their difference lay and how does this make both necessary? I am not saying that these are identical and thus redundant, but I do struggle to understand how they differ and in what their uniqueness lay.
> >
> > **3.** What is $\hat{\mathbf{a}}$? How is it computed?

---

> ### Author Response · Authors · 2025-11-25
>
> We thank the reviewer for the continued engagement and the opportunity to clarify these subtle but critical mechanisms. We understand your concern: if all networks are learning simultaneously, how do we avoid the "chicken-and-egg" problem?
>
> Below, we provide a detailed explanation of why POW fundamentally differs from the heuristic approach of WQMIX and how the architecture ensures convergence.
>
> ### 1. Why POW avoids the "Gap" of WQMIX and ensures correct convergence.
>
> You asked if learning $\hat{Q}^*$ and $Q_r$ simultaneously risks converging to a wrong equilibrium, similar to WQMIX. The short answer is: **No, because POW constructs a mathematically valid Bellman optimality operator, whereas WQMIX relies on inconsistent heuristics.**
>
> Here is the step-by-step logic:
>
> * **The Goal of Weighted Training:**
>     To ensure $\hat{Q}^\*$ converges to the true $Q^\*$, the training target must be the Bellman Optimality Equation: $y = r + \gamma \max_{\boldsymbol{a}'} \hat{Q}^\*(\tau', \boldsymbol{a}')$.
>     In value decomposition, we use $Q_{tot}$ to select actions. Thus, we need the condition: $\arg\max Q_{tot} = \arg\max \hat{Q}^\*$. If this condition holds, the target becomes $y = r + \gamma \hat{Q}^*(\tau', \arg\max Q_{tot})$, which is the valid optimality target.
>
> * **The Gap in  WQMIX:**
>     WQMIX *assumes* it can assign higher weights to the optimal actions of $\hat{Q}^\*$ to force $Q_{tot}$ to align with it. However, finding $\arg\max \hat{Q}^\*$ is computationally intractable. Therefore, WQMIX uses **heuristics** (e.g., comparing $y$ vs. $Q_{tot}$ or $\hat{Q}^*$) to decide which actions are optimal (please refer to Eqn. 10 in WQMIX paper). These heuristics are often inaccurate during training, leading to the gap you correctly identified.
>
> * **The Solution of POW (Bridging the Gap):**
>     We do not use heuristics. Instead, we train $Q_r$ to explicitly recognize the potentially optimal actions of the **current** $\hat{Q}^*$.
>     1. We do not require $\hat{Q}^\*$ to equal the true $Q^\*$ immediately. We only require $Q_r$ to identify the high-value actions of the *current* estimate $\hat{Q}^\*$. Since they share the same objective, $Q_r$ effectively tracks $\hat{Q}^\*$.
>     2. By up-weighting actions in $A_r$ (identified by $Q_r$), we force $Q_{tot}$ to verify: $\arg\max Q_{tot} \approx \arg\max \hat{Q}^\*$.
>     3. Once $Q_{tot}$ is aligned with the current $\hat{Q}^\*$, the update target for $\hat{Q}^\*$ becomes the valid Bellman Optimality target. As proven in classic RL theory, applying the valid Bellman operator repeatedly guarantees that $\hat{Q}^\*$ converges to the true $Q^\*$.
>
> In summary, we do not need to "trust" $\hat{Q}^\*$ to be correct initially. We only ensure the mechanism ($Q_r$) consistently aligns $Q_{tot}$ with $\hat{Q}^\*$. This creates a stable contraction mapping towards $Q^\*$, unlike WQMIX where heuristic errors break this loop.
>
> ### 2. Why is $Q_r$ necessary?
>
> The distinction between $Q_r$ and $\hat{Q}^\*$ lies entirely in **Computational Complexity** and **Architecture constraints regarding Action Selection**.
>
> * **$\hat{Q}^\*$ (Unrestricted, Intractable Maximization):**
>     $\hat{Q}^\*$ is intended to approximate the true value. It has **no structural constraints**, which allows it to represent any value function. However, finding its greedy action $\arg\max_{\boldsymbol{a}} \hat{Q}^\*(\tau, \boldsymbol{a})$ requires iterating over the entire joint action space (Exponential Complexity, $|A|^N$), which is impossible during training.
>
> * **$Q_r$ (Unrestricted Value, Tractable Maximization via IGM):**
>     $Q_r$ is designed with a specific architecture with joint action $\boldsymbol{a}$ as input and modeled by $\lambda (Q_i - max Q_i)$ (**please refer to Eqn. 6 in the paper**), making it **unconstrained by monotonicity but satisfies the IGM condition**. This architectural constraint grants us a capability that $\hat Q^\*$  lacks: Linear-time Maximization.
>
> Therefore, because $Q_r$ satisfies IGM, we can determine if a joint action $\boldsymbol{a}$ is "potentially optimal" by comparing its value $Q_r(\tau, \boldsymbol{a})$ against the greedy baseline $Q_r(\tau, \hat{\boldsymbol{a}})$. Thanks to the IGM property of $Q_r$, calculating $Q_r(\tau, \hat{\boldsymbol{a}})$ has **Linear Complexity** ($O(N)$).
>
>
> ### 3. Definition of $\hat{\mathbf{a}}$
>
> $\hat{\mathbf{a}}$ represents the **decentralized greedy joint action**.
>
> Since $Q_{tot}$ and $Q_r$ are designed to satisfy the IGM condition, the global greedy action is simply the collection of each agent's local greedy action. Mathematically:
>
> $$\hat{\boldsymbol{a}} = \arg\max_{\boldsymbol{a}} Q_{tot}(\tau, \boldsymbol{a}) = \left[ \arg\max_{a_1} Q_1(\tau_1, a_1), \dots, \arg\max_{a_n} Q_n(\tau_n, a_n) \right]$$
>
> In our implementation, $Q_{tot}$ and $Q_r$ share the same individual utility functions $Q_i$. Therefore, $\hat{\mathbf{a}}$ is computed efficiently by each agent selecting its local best action, avoiding any global search.

---

> > ### Comment · Reviewer_LRKB · 2025-11-25
> > **Reply to Authors**
> >
> > I would like to once again thank the authors for their detailed comments, which helped me to improve my understanding of the paper. I appreciate the explanation given in the last comment, this makes it clear some aspects that did not come through the paper immediately. I really suggest to revise the explanation of the working mechanism of POW in the paper to follow such an explanation, as I think it would prove useful for all readers generally. I raised my score to account for the authors clarifications that addressed my concerns.

---

> > > ### Author Response · Authors · 2025-11-25
> > >
> > > We sincerely thank the reviewer once again for the constructive feedback and for improving the score. We will revise the paper accordingly to make our contributions clearer and to better address the reviewer’s concerns.

---

### Official Review · Reviewer_Nw5q · 2025-10-30

**Soundness:** 3
**Presentation:** 3
**Contribution:** 3
**Rating:** 6
**Confidence:** 3

**Summary:**

This paper proposes POW (Potentially Optimal Joint Actions Weighting), a value-decomposition method for cooperative MARL. It introduces a recognition module $Q_r$ that identifies potentially optimal joint actions and up-weights them during training. The authors prove convergence guarantees and evaluate POW on matrix games, predator–prey, SMAC, and other benchmarks, demonstrating improved performance over QMIX, WQMIX variants, and other MARL algorithms.

**Strengths:**

1. The paper addresses the gap between WQMIX theory and its heuristic approximations.
2. The authors provide proof that the recognized action set converges to optimal actions.
3. The method was tested on extensive benchmarks, showing performance gain across both monotonic and non-monotonic tasks.

**Weaknesses:**

See the questions section.

**Questions:**

Questions about the current manuscript:
1. Will $Q_r$ and iterative loops increase implementation difficulty compared to QMIX or QPLEX? If so, how can this be mitigated?
2. May not be strictly required, but I wonder if any possible comparisons with CTDE actor-critic MARL methods like MAPPO?
3. Can we include the algorithm pseudo-code for clarity?
4. [Minor]: Line 423: the cited work REMIX seems not to match the given reference. Please check.

---

> ### Author Response · Authors · 2025-11-23
>
> ## **Response to Questions**
>
> **1. On Implementation Difficulty**
>
> Although $Q_r$ is trained iteratively, it does not increase the implementation difficulty relative to QMIX or QPLEX. The key reason is that $Q_r$ satisfies the IGM property (without monotonicity). **This allows $Q_r$ to efficiently evaluate whether a joint action is potentially optimal by simply comparing $Q_r(\tau,\boldsymbol{a})$ with the baseline $Q_r(\tau,\hat{\boldsymbol{a}})$, computed in linear time, without enumerating the joint action space**. Therefore, POW avoids the exponential complexity that would otherwise increase implementation difficulty.
>
> We also evaluated actual training time on a single RTX 3090. In Predator-Prey, QMIX and POW-QMIX both take about 14 hours (QPLEX 12 h, ResQ 23 h). In 3s5z_vs_3s6z, QMIX takes 11 hours, POW-QMIX takes 12, QPLEX 10.5, and ResQ 15 hours. Thus, **POW introduces no practical overhead and is substantially lighter than ResQ-style full-space weighting**.
>
> Finally, increasing the network size of WQMIX to match POW-QMIX still fails to recover performance, demonstrating that our improvement does not come from heavier computation or larger models, but from correctly identifying optimal actions in non-monotonic tasks.
>
>
> **2. On Comparison with Actor-Critic Methods like MAPPO**
>
> This is an excellent suggestion. As stated in the related work section, the primary focus of this paper is to fix the gap between theory and practical implementation within the paradigm of weighted value-decomposition methods, e.g. WQMIX.
>
> While MAPPO and other actor-critic algorithms are indeed state-of-the-art in MARL, they represent a different algorithmic family. A comparison would be interesting but is largely orthogonal to our core research question. We thank you for the suggestion and will certainly consider including a wider range of baselines, including prominent actor-critic methods, in our future work.
>
> **3. On Including Pseudocode**
>
> Certainly. We agree that including pseudocode would significantly improve the clarity of our method's implementation. **We have added a detailed algorithm block in the revised manuscript, please see Section 3.4**.
>
> **4. On the Citation for REMIX**
>
> Thank you very much for catching this error. We sincerely apologize for the oversight and **have corrected the reference for REMIX in the revised paper**.

---

### Official Review · Reviewer_P6qW · 2025-10-31

**Soundness:** 3
**Presentation:** 3
**Contribution:** 2
**Rating:** 2
**Confidence:** 4

**Summary:**

This paper proposes a new method named Potentially Optimal Joint Actions Weighting (POW) to address the problem of representativeness of a wide range of value factorization functions. This is achieved by assigning higher weights to actions that are found with potential to be optimal.

**Strengths:**

This work good theoretical groundings and a comprehensive range of experiments across different MARL environments. The matrix games provided are also useful to demonstrate the representational properties of the method and support the claims made. Theorem 1 and definition 1 are sound.

**Weaknesses:**

A number of different methods for value function factorization has been explored recently, and some of them can theoretically guarantee the factorization of any family of the environments (such as QTRAN). I feel the motivations for the proposed method are hence not sufficient, simply by saying that their object is to try to improve the expressiveness of the range of factorisation functions that can be represented.

In section 4.5.2, WQMIX seems to be better than POW in Figure 7(a) and 7(b); it is unclear to me how these results show that the performance of POW "stems from its recognition-weighting design, not from parameter count" (line 400); what it shows is that other methods such as WQMIX improve when the network size increases and QPLEX stays the same; it is unclear how it says something about POW's parameter count and in fact it shows that WQMIX performs better in 7(a) and 7(b) when the number of parameters is the same.

Some notations could also be made more clear; for instance, in figure 1, it is unclear to me the meaning of $A_{tot}$.

Please find below some more questions that reflect other concerns.

**Questions:**

1. in lines (168-169): " In all our experiments, we set $\alpha=0$, so only actions in $A_r$ contribute to updates, aligning theory with practice." - so what is the pointof proposing this $\alpha$ weight?
2. i wonder what happens if $\alpha$ is not 0? has that been tested?
3. could the authors elaborate on what is the loss "$L_{Q^*}$" in figure 1?
4. in eq 4, is the optimal value function $Q_*$ also learned? or is it known a priori? from the notation it seems almost like an on policy approach being described, which makes the link between notation and practice a bit unclear
5. in figure 3, the performances across the 3 different levels of penalties seem a bit inconsistent; for example, could the authors elaborate on why the proposed method performs better in p=-4 and p=-5 but not in p=-3?

---

> ### Author Response · Authors · 2025-11-23
>
> ## **Response to Weaknesses**
>
> **1. On the Motivation of the Proposed Method**
>
> We agree that methods like QPLEX and QTRAN can theoretically guarantee the factorization of any environment. But as shown in our matrix game experiments, **methods like QPLEX, despite their representational power, can fail to converge even in relatively simple settings due to the lack of an effective weighting mechanism** to guide the learning process.
>
> On the other hand, methods with weighting like **WQMIX suffer from a significant gap** between their theoretical ideal (weighting based on the true optimal action) and their practical implementation (using heuristics to approximate it), which limits their performance.
>
> Our work, **POW, is the first to bridge this divide**. It unifies the full factorization capability with a theoretically sound and practically realizable weighting mechanism. By doing so, POW ensures both representational completeness and implementation consistency.
>
> **2. On the Ablation Study and Parameter Count**
>
> We appreciate you pointing out the details in Figure 7. The primary purpose of this experiment is to demonstrate that POW's superior performance stems from its recognition-weighting design, not simply from a higher parameter count.
>
> *   **The key evidence lies in Figure 7(c) (corridor environment)**. As shown, even when WQMIX's network size is increased to match that of POW-QMIX, it still fails to learn. **This strongly indicates that the performance gain in complex, non-monotonic environments comes from our method's ability to correctly identify and focus on optimal actions**, a capability that simply adding more parameters to WQMIX does not confer.
> *   **In the simpler environments of Figure 7(a) and 7(b)**, both methods successfully converge to the optimal policy. **The increased parameter count does grant WQMIX a faster convergence speed in these cases, but the final performance is the same**. This does not contradict our central claim that POW's design enables robust convergence and stability across a wider range of environments, especially those where other methods fail.
>
> **3. On the Notation of $A\_{tot}$**
>
> The notation $A_{tot}$ in Figure 1 refers to the mixing of the individual agent advantage functions. We adopted this notation from QPLEX to maintain consistency with related literature in the field. We have provided more detailed notation definitions in the revision, please see Section 3.1.
>
> ## **Response to Questions**
>
> **1. On the Purpose of the $\alpha$ Weight**
>
> We introduced **the $\alpha$ weight primarily to maintain a parallel structure with WQMIX's weighting formulation**, which makes the conceptual comparison between the two methods clearer. **We set $\alpha=0$ in our experiments to ensure a perfect alignment between our theory and practice**, guaranteeing that only the identified potentially optimal joint actions contribute to the updates. **In contrast, WQMIX's $\alpha$ is a sensitive hyperparameter** that, if not tuned correctly, can further widen its theory-practice gap.
>
> **2. On the Loss "$L_{\hat{Q}^\*}$" in Figure 1**
>
> $L\_{Q^\*}$ in Figure 1 is the loss for training $\hat{Q}^\*$ (**Equation 4a in the revised version**). The training target is $y$ as described in **Equation 5**.
>
> **3. On Whether $Q^\*$ is Learned**
>
> We have proved that the iterative weighted training mechanism theoretically ensures that with our recognition network, $Q_r$, $\hat{Q}^\*$ converges towards the true (but initially unknown) $Q^\*$ over the course of training.
> The descriptions of $Q_r$, $\hat{Q}^\*$ and $Q^\*$ are provided in the general response.
>
> **4. On Performance Inconsistency in Predator-Prey**
>
> The performance across the different penalty levels **is consistent in terms of the final outcome and general trend**. POW-QMIX successfully converges to the optimal policy in all three settings ($p=-3, -4, -5$), with the convergence speed naturally decreasing as the task difficulty increases.
>
> In the easiest case ($p=-3$), OW-QMIX converges faster. This is likely because its simple optimistic weighting heuristic is sufficient for this lower difficulty level. However, this same heuristic fails in the more challenging $p=-4$ and $p=-5$ settings. In contrast, our method remains robust and effective across all difficulty levels, highlighting its superior stability and reliability.
>
> Thank you once more for your valuable feedback. Please let us know if there are any further questions or points we can clarify.

---

> > ### Author Response · Authors · 2025-11-25
> >
> > We would like to offer one additional clarification regarding the role of **asymptotic performance** in reinforcement learning.
> >
> > In RL, the learning process is inherently exploratory, and **early-stage fluctuations or slower initial improvement do not reflect the capability of a method to ultimately acquire a high-quality policy**. What fundamentally characterizes an RL algorithm is **its fixed point—i.e., the policy it converges to once learning stabilizes**. This long-term performance determines whether the method can truly solve the underlying decision-making problem, especially in settings where exploration and credit assignment are difficult.

---

### Official Review · Reviewer_ZhCF · 2025-11-01

**Soundness:** 2
**Presentation:** 3
**Contribution:** 3
**Rating:** 6
**Confidence:** 4

**Summary:**

The paper proposes POW, a value decomposition based MARL method under CTDE for optimal joint policy recovery using recognition weighting design. Empirically, the method demonstrates improved performance over baselines on multiple benchmarks

**Strengths:**

- This paper is generally well written and easy to follow; With a constrained Q_r to over include promising joint actions and focus learning signal on them and bridges the gap of WQMIX in ideal weighting and practical approximation.

- There are multiple benchmarks demonstrating the effectiveness empirically.

**Weaknesses:**

- Some of the claims can be too strong. For theorem 1, it assumes the condition of Q_r can converge to  ${Q^\star}$, however  ${Q^\star}$ is also unknown and to be learned. E.g in eq.4 it is not using \hat{Q^\star} but ${Q^\star}$, yet also claims Q_r is optimizized to $\hat{Q^\star}$. The gurantee in theorem 2 also requires A_r to converge to only optimal a; These are the actual challenging points in practice and cannot be just assumed; This setting is not beneficial to the actual experiment or pracitcal settings.

- Also as above, Q_r and Q_tot rely on $ \hat{Q^\star}$ for bootstrapping, however how the bias in  $\hat{Q^*}$ will effect A_r and Q_r is not discussed. Under extrem case they will not converge at all.

- Some of the SOTA works are missing. This paper only compares with some of the classic Cooperative MARL works.

**Questions:**

- I believe in multiple equations $ \hat{Q^\star}$ and  ${Q^\star}$ and mixed and used against the verbal description in paper, making it hard to understand if it's typo, approximation or error.

- Q_r(s,a) and Q_r (\tau, a) are mixed and used; According to verbal description it should be Q_r(s,a)?

- I wonder the C in eq5 can be discussed in terms of its sensitivity

- Is Q^* and Q_tot network the same as in WQMIX? Also I wonder the \alpha setting used for WQMIX in the experimental setting

---

> ### Author Response · Authors · 2025-11-23
>
> ## **Response to Weaknesses**
>
> **1. On the Convergence Guarantees of $Q_r$ and $A_r$**
>
> Please refer to our general response.
>
> **2. On the Choice of Baselines and Missing SOTA Works**
>
> We carefully selected our baselines to ensure a fair and meaningful comparison. We chose **QMIX, WQMIX, QPLEX, and ResQ** because they are classic, highly representative, and, most importantly, **closely related to the core focus of our work**: addressing the suboptimality in value decomposition methods through potentially optimal joint action recognition and weighted learning.
>
> Some more recent SOTA methods, such as CIA[1] or transformer-based approaches (e.g., VDT[2]), primarily enhance MARL performance by incorporating temporal information or employing more complex network architectures. While effective, their contributions are largely **orthogonal** to our paper's central theme of fixing the gap between theory and implementation in weighted training. Therefore, comparing against them would not directly evaluate the specific problem we aim to solve.
>
> Nevertheless, we have enriched the discussion of these methods in related works.
> ```
> [1] Liu, S., Zhou, Y., Song, J., Zheng, T., Chen, K., Zhu, T., ... & Song, M. (2023). Contrastive identity-aware learning for multi-agent value decomposition. In Proceedings of the AAAI Conference on Artificial Intelligence (Vol. 37, No. 10, pp. 11595-11603).
> [2] Zhao, Z., Zhang, Y., Chen, W., Zhang, F., Wang, S., & Zhou, Y. (2025). Sequence value decomposition transformer for cooperative multi-agent reinforcement learning. Information Sciences, 122514.
> ```
>
> ## **Response to Questions**
>
> **1. On the Mixed Use of $Q^\*$ and $\hat{Q}^\*$**
>
> We apologize for the confusion caused by the notation. Their definitions are provided in the general response. We have revised the manuscript to clearly describe the notations, please see Section 3.1.
>
> **2. On the Mixed Use of $Q_r(s, a)$ and $Q_r(\tau, a)$**
>
> You are correct! We apologize for the inconsistent notations.
> In fact $Q_r(\tau, a)$ is a deliberate notational simplification for conciseness and consistency with prior work:
> *   In our framework (as in QPLEX, ResQ), the global state $s$ is theoretically required and is used to construct the hypernetwork for both $Q_r$ and $Q_{tot}$.
> *   However, from an individual agent's perspective, the input is its local action-observation history $\tau$.
> *   Therefore, we choose to use the simplified notation $Q_r(\tau, a)$ to represent the network's output, where the dependence on $s$ is implicitly handled by the hypernetwork.
>
> We have consistently replaced $Q_r(s, a)$ with $Q_r(\tau, a)$ to avoid ambiguity.
>
> **3. On the Sensitivity of Constant $C$**
>
> This is an excellent question. Theoretically, $C=0$ is ideal. However, in practice, due to function approximation and optimization errors, the $Q_r$ estimates may have slight inaccuracies. To enhance the stability of identifying the set of potentially optimal joint actions ($A_r$), we introduce $C$ as a small positive constant. This provides a tolerance margin, preventing true optimal actions from being mistakenly excluded due to minor estimation noise.
> **The setting of $C$ values across all environments are reported in the Appendix F (0.05 for Predator-prey, SMAC and SMAC v2, 0.1 for Highway-env)**. As shown, the values are almost consistent across different tasks. The empirical results indicate that the model's convergence and performance are not highly sensitive to the precise value of $C$.
>
> **4. On Network Structures and $\alpha$ Setting for WQMIX**
>
> * **Network Structure**: Yes, the network architecture for $Q_{tot}$ and $\hat{Q}^\*$ is identical to that in WQMIX. Our contribution lies in introducing the additional recognition network **$Q_r$** and the **iterative weighted training mechanism** by the recognition of potentially optimal joint actions, not in altering the base network structure.
> *   **$\alpha$ Setting for WQMIX**: For the WQMIX baseline, we used the default value of **$\alpha=0.1$** as recommended in the original WQMIX paper for our experiments.
>
> Thank you again. We are happy to provide further clarification if any additional questions arise.

---

### Author Response · Authors · 2025-11-23
**General Response**

We sincerely thank all reviewers for their time and insightful comments. Here we provide a consolidated overview of our core design philosophy, which we believe will help address several common points.

### 1. $Q^\*$, $\hat{Q^\*}$, $Q^{tot}$, $Q_r$ and their roles in POW.

• $Q^\*$ represents the **true (but unknown) optimal joint action value function**. We use $Q^\*$ primarily in our theoretical analysis and proofs to denote the ultimate convergence target.

• $\hat{Q}^\*$ represents the estimated optimal joint action value function used during the practical training process, providing an **unrestricted bootstrap target**.

• $Q_{tot}$ satisfies the Individual-Global-Max (IGM) condition and monotonicity constraint  to select the optimal joint action **for decentralized execution**.

• $Q_r$ **takes the joint action as input** and provides an expressive joint action value model **unconstrained by monotonicity but conforming to IGM** (see Eqn.6 in the revised paper and its explanation). Its function is to **recognize whether $\boldsymbol{a} \in A_r$** or not.

Therefore, $\hat{Q^\*}$, $Q^{tot}$, $Q_r$ are not interchangeable because they serve different and complementary roles in POW. However, they share the same TD target so that action selection and value estimation remain consistent across components and learning remains stable:

$\mathcal{L}_{\hat Q^\*} = \mathbb{E}[(\hat Q^\*(\tau,a)-y)^2]$

$\mathcal{L}\_{Q\_{tot}} = \mathbb{E}[w(s,a),(Q_{tot}(\tau,a)-y)^2]$

$\mathcal{L}_{Q_r} = \mathbb{E}[(Q_r(\tau,a)-y)^2]$

$y = r + \gamma \hat{Q}^\*(\tau', \arg \max_{\boldsymbol{a}} Q_{tot}(\tau', \boldsymbol{a}))$

Our **Iterative Weighted Training** process is designed to ensure that $\hat{Q}^\*$ is provably guided to converge towards the true $Q^\*$.

### 2. The rationale of $Q_r$ and Iterative Weighted Training
POW addresses a fundamental challenge in value decomposition methods: the discrepancy between the theoretical need to identify optimal joint actions and the practical difficulty of exhaustively searching over all joint actions. WQMIX proposes weighting but relies on heuristics or approximations in practice, creating a gap between their theoretical guarantees and implementation.

To solve this, we introduce the $Q_r$ module.

1.  **To identify potentially optimal joint actions**, each joint action must be evaluated independently. Therefore, $Q_r$ takes the joint action $\boldsymbol{a}$ as an explicit input and is free from the monotonicity constraints.

2.  **To avoid searching the entire joint action space**, we need an efficient benchmark. We achieve this by comparing any action's value $Q_r(s, \boldsymbol{a})$ against the value of the decentralized greedy action, $Q_r(s, \hat{\boldsymbol{a}})$, which has only linear complexity with respect to the number of agents.

3. **Our iterative weighted training mechanism ensures that the set of recognized actions converges to the true optimal set, and that $Q_{tot}$ converges to the optimal policy**. These guarantees are formally provided in Theorem 1 and Theorem 2, with detailed proofs in the appendix.

We believe this overview clarifies the foundational principles of our work. The manuscript has also been revised accordingly.

---

### Meta-Review · Area_Chair_G2uM · 2026-01-03

**Summary:**

This work proposes POW-QMIX, a Potentially Optimal Joint Actions Weighting method for Multi-Agent Reinforcement Learning Value Factorization. It overcomes the limitations of Weighted QMIX, which relies on heuristics to find optimal actions. POW-QMIX works by introducing $Q_r$ to identify the set of potential optimal actions, and then weights these actions during learning. It shows theoretically that their method could find the optimal actions if $Q_r$ converges.

The strengths of this work are summarized as follows.
1. The potential weighting method could overcome the drawback of Weighted QMIX.
2. The theoretical results show that POW-QMIX could converge to the optimal actions.
3. The performance of this work is promising.


The weaknesses of this work are summarized as follows.

1. This work is built on the assumption that $Q_r$ converges. However, it is not guaranteed.
2. Albeit, $Q_r$ can obtain the set of potential optimal actions. Using the Bellman optimality operator based on the potential optimal actions (not the true optimal actions of the current policy) does not guarantee  $\hat{Q*}$  converges to the true $Q*$. It is unclear whether the contraction mapping can still be found.



Please discuss the weakness in the main text if this work is accepted to appear in the conference.

**Reviewer Concerns:**

The reviewers were concerned with the motivation, notations, the loss function, the convergence of this propose method.

**Reviewer Scores:**

One reviewer increased the rating from 4 to 6. One reviewer (rating 2) were concerned with the motivation, the performance, and the notations. The authors have addressed the concerns of the reviewer.

---

### Decision · Program_Chairs · 2026-01-26

Accept (Poster)